# Maternal-fetal immune responses in pregnant women infected with SARS-CoV-2

Valeria Garcia-Flores[1,2], Roberto Romero [1,3,4,5,6✉], Yi Xu[1,2], Kevin R. Theis[1,2,7], Marcia Arenas-Hernandez [1,2], Derek Miller [1,2], Azam Peyvandipour[1,2,5], Gaurav Bhatti[1,2], Jose Galaz[1,2], Meyer Gershater[1,2], Dustyn Levenson[1,2], Errile Pusod[1,2], Li Tao[1,2], David Kracht[1,2], Violetta Florova[1,2], Yaozhu Leng[1,2], Kenichiro Motomura [1,2], Robert Para[1,2], Megan Faucett[1,2], Chaur-Dong Hsu[1,2,8], Gary Zhang [5], Adi L. Tarca [1,2,9], Roger Pique-Regi[1,2,5✉] & Nardhy Gomez-Lopez [1,2,7✉]

Pregnant women represent a high-risk population for severe/critical COVID-19 and mortality. However, the maternal-fetal immune responses initiated by SARS-CoV-2 infection, and whether this virus is detectable in the placenta, are still under investigation. Here we show that SARS-CoV-2 infection during pregnancy primarily induces unique inflammatory responses at the maternal-fetal interface, which are largely governed by maternal T cells and fetal stromal cells. SARS-CoV-2 infection during pregnancy is also associated with humoral and cellular immune responses in the maternal blood, as well as with a mild cytokine response in the neonatal circulation (i.e., umbilical cord blood), without compromising the T-cell repertoire or initiating IgM responses. Importantly, SARS-CoV-2 is not detected in the placental tissues, nor is the sterility of the placenta compromised by maternal viral infection. This study provides insight into the maternal-fetal immune responses triggered by SARS-CoV-2 and emphasizes the rarity of placental infection.

[1] Perinatology Research Branch, Division of Obstetrics and Maternal–Fetal Medicine, Division of Intramural Research, Eunice Kennedy Shriver National Institute of Child Health and Human Development, National Institutes of Health, US Department of Health and Human Services (NICHD/NIH/DHHS), Bethesda, MD 20892 and Detroit MI 48201, USA. [2] Department of Obstetrics and Gynecology, Wayne State University School of Medicine, Detroit, MI 48201, USA. [3] Department of Obstetrics and Gynecology, University of Michigan, Ann Arbor, MI 48109, USA. [4] Department of Epidemiology and Biostatistics, Michigan State University, East Lansing, MI 48824, USA. [5] Center for Molecular Medicine and Genetics, Wayne State University, Detroit, MI 48201, USA. [6] Detroit Medical Center, Detroit, MI 48201, USA. [7] Department of Biochemistry, Microbiology and Immunology, Wayne State University School of Medicine, Detroit, MI 48201, USA. [8] Department of Physiology, Wayne State University School of Medicine, Detroit, MI 48201, USA. [9] Department of Computer Science, Wayne State University College of Engineering, Detroit, MI 48201, USA. ✉email: prbchiefstaff@med.wayne.edu; rpique@wayne.edu; nardhy.gomez-lopez@wayne.edu

To date, more than 150,000 pregnant women in the United States have been infected with SARS-CoV-2[1], the virus responsible for the coronavirus disease 2019 (COVID-19). During pregnancy, SARS-CoV-2 infection can lead to variable outcomes, which range from experiencing no symptoms to developing severe/critical disease[2,3]. Most pregnant women with SARS-CoV-2 infection are asymptomatic or only experience mild symptoms[4,5]. Regardless, in the first 6 months of the COVID-19 pandemic, it was documented that pregnant women with SARS-CoV-2 were at an increased risk for hospitalization, mechanical ventilation, intensive care unit admission, and preterm birth[2,3,6,7], but rates of maternal mortality were reported to be similar between pregnant and non-pregnant women[6]. More recently, it has been clearly shown that pregnant women are at high risk for severe/critical disease and mortality as well as preterm birth[8–11]. Therefore, investigating host immune responses in pregnant women infected with SARS-CoV-2, even if they are asymptomatic, is timely.

Most neonates born to infected women test negative for SARS-CoV-2, and the majority of those testing positive for the virus present symptoms that are not severe[7,12]. For the latter group, the timing of mother-to-child transmission (i.e., vertical transmission) of SARS-CoV-2 is still unclear, since this can occur in utero, intrapartum, or early in the postnatal period[13]. Yet, while rare[13], there is already evidence of SARS-CoV-2 in utero vertical transmission[14,15], which is likely to occur through the hematogenous route (i.e., bloodstream infection)[16]. In such cases, the virus must cross the maternal–fetal interface by infecting the syncytiotrophoblast layer of the placenta to gain access to the fetal circulation. The mechanisms whereby SARS-CoV-2 infects placental cells are still under investigation; however, it is well accepted that coronaviruses can enter host cells via two main canonical mechanisms[17,18]: (1) the direct pathway, in which host cells are required to express both the angiotensin-converting enzyme 2 (ACE-2) receptor[19] and the serine protease TMPRSS2[20]; and (2) the endosomal route, in which cell entry can be mediated by ACE-2 alone. Using both single-cell and single-nuclear RNA sequencing, we have previously shown that the co-expression of ACE-2 and TMPRSS2 is negligible in first, second, and third trimester placental cells[21]. Subsequent investigations demonstrated that the ACE-2 protein was polarized to the stromal (fetal) side of the syncytiotrophoblast and TMPRSS2 was limited to the villous endothelium[22,23]. Yet, placental cells can express non-canonical cell entry mediators such as cathepsin L (CSTL), FURIN, and sialic acid-binding Ig-like lectin 1 (SIGLEC1), among others[21]. Furthermore, SARS-CoV-2 infection can be associated with vascular damage in pregnant women, in whom ischemic injury of the placenta may facilitate viral cell entry[24]. Therefore, SARS-CoV-2 can infect placental cells, as has already been reported[25]; however, placental infection alone is not considered confirmatory evidence of in utero vertical transmission[13]. Nonetheless, it is possible that the maternal inflammatory response induced by SARS-CoV-2 infection has deleterious effects on the offspring. Therefore, investigating the host immune response in the umbilical cord blood as well as at the site of maternal–fetal interactions (i.e., the maternal–fetal interface) may shed light on the adverse effects of SARS-CoV-2 infection during pregnancy.

Herein, we utilize a multidisciplinary approach that includes the detection of SARS-CoV-2 IgM/IgG, multiplex cytokine assays, immunophenotyping, single-cell RNA-sequencing (scRNA-seq), bulk transcriptomics, and viral RNA and protein detection, together with the assessment of the microbiome diversity and histopathology of the placenta, to characterize the maternal–fetal immune responses triggered by SARS-CoV-2 during pregnancy. We report that SARS-CoV-2 during pregnancy initiates unique maternal and fetal immune responses in the maternal and neonatal circulation as well as at the maternal–fetal interface in the absence of viral detection in the placenta. This study highlights the deleterious effects of SARS-CoV-2 infection during pregnancy on the mother and the offspring.

## Results

**Characteristics of the study population**. A total of 23 pregnant women were enrolled in our study. The demographic and clinical characteristics of the study population are displayed in Supplementary Table 1. Maternal blood samples were collected upon admission, prior to administration of any medication. Twelve pregnant women tested real-time polymerase chain reaction (RT-PCR) positive (nasopharyngeal swab) for SARS-CoV-2; eight were asymptomatic, one had mild symptoms (e.g., fever and tachycardia), and three were diagnosed as having severe COVID-19 (requiring oxygen supplementation). One of the women with severe disease underwent emergency preterm cesarean section due to worsening respiratory function, which is consistent with previous studies reporting that COVID-19 is associated with higher rates of indicated preterm birth[10]. Yet, the rest of the SARS-CoV-2-positive women delivered term neonates, as did most of the non-infected controls. Neonates were not RT-PCR tested for SARS-CoV-2; thus, infection status throughout the manuscript refers solely to the mother. No differences in demographic and clinical characteristics were found between the study groups, including Apgar scores and placental histopathological lesions.

**Pregnant women with SARS-CoV-2 infection and their neonates exhibit distinct IgM responses**. Previous studies have shown that maternal IgG antibodies are transferred across the placenta in both symptomatic and asymptomatic women infected with SARS-CoV-2[26]. In addition, there is evidence showing that neonates born to mothers with COVID-19 can have detectable SARS-CoV-2 IgM as well as IgG[15,27]. The presence of IgG is likely due to the passive transfer of this immunoglobulin from the mother to the fetus across the placenta. However, detectable levels of IgM suggest that the fetus was infected with SARS-CoV-2, given that this immunoglobulin cannot cross the placenta due to its large molecular weight. Therefore, we first determined the concentrations of SARS-CoV-2-specific IgM and IgG in the maternal and umbilical cord blood (hereafter referred to as "cord blood"). As expected, pregnant women with SARS-CoV-2 infection had higher serum levels of IgM and IgG than controls (Fig. 1a). The IgM and IgG levels of the pregnant women with severe COVID-19 were similar to those without symptoms. In addition, IgG was increased in the cord blood of neonates born to women infected with SARS-CoV-2 infection but IgM was undetected, similar to control neonates (Fig. 1a). Therefore, serological data imply that in our study population, which is largely asymptomatic for COVID-19, none of the neonates seemed to be infected with SARS-CoV-2.

**Pro-inflammatory cytokine responses are displayed in the circulation of pregnant women with SARS-CoV-2 infection and their neonates**. The pathophysiology of SARS-CoV-2 infection includes a cytokine storm in the systemic circulation, which can lead to multi-organ damage[28,29]. Hence, we next determined the systemic cytokine response in mothers and neonates by measuring the concentrations of 20 cytokines in maternal and cord blood plasma. Pregnant women infected with SARS-CoV-2 had increased systemic concentrations of IL-8 (5.9-fold change (FC)), IL-10 (2.3-FC), and IL-15 (1.5-FC) compared to control mothers (Fig. 1b, Supplementary Table 2); such changes were not driven solely by the severe COVID-19 cases. Neonates born to women

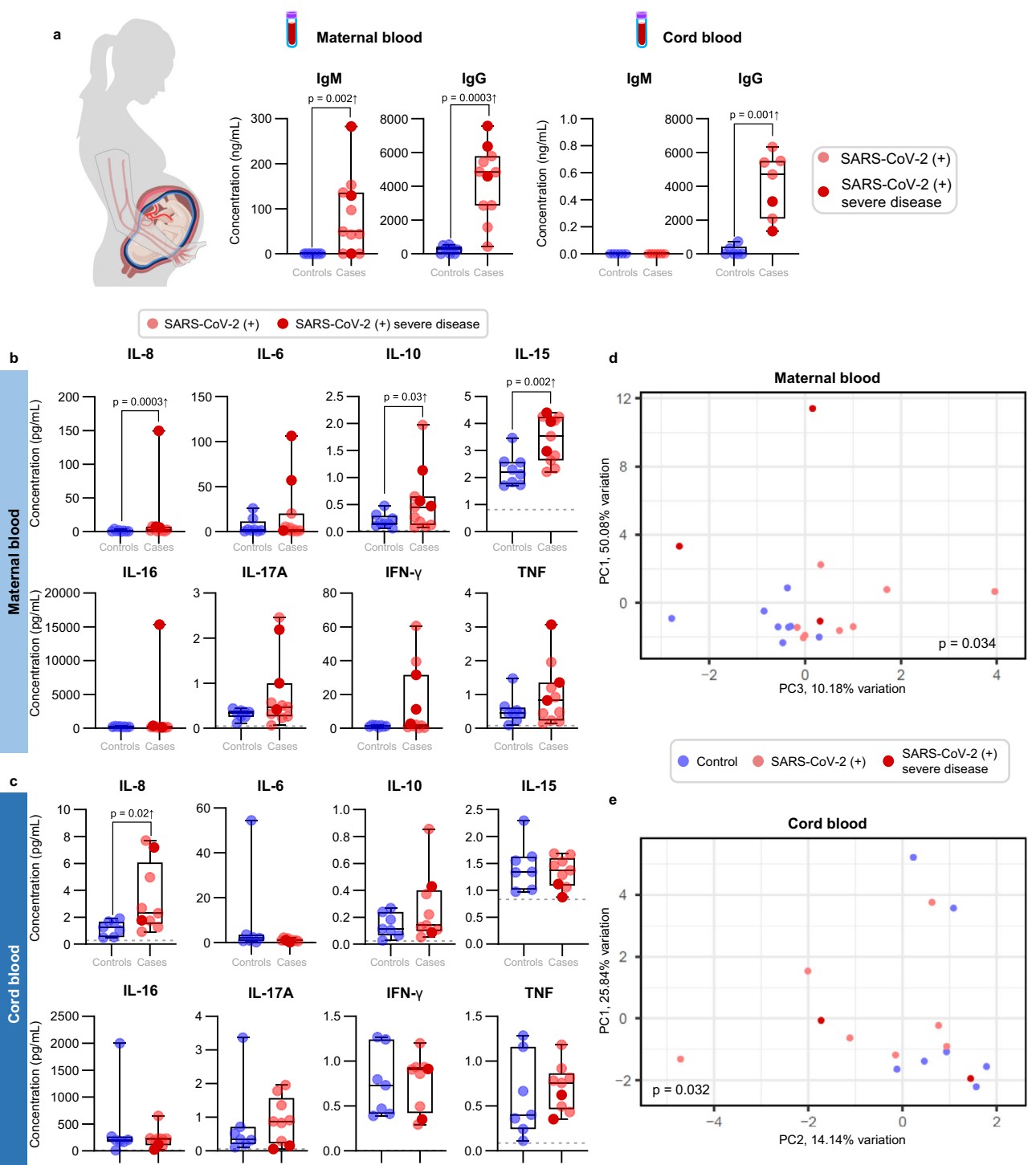

**Fig. 1 Serological and cytokine responses in maternal and cord blood of women with SARS-CoV-2 infection and their neonates. a** Serum concentrations of IgM and IgG in the maternal blood [$n = 9$ control, 11 SARS-CoV-2 (+) (left panel) and cord blood [$n = 6$ control, 7 SARS-CoV-2 (+)] (right panel). Data are shown as boxplots where midlines indicate medians, boxes indicate interquartile range and whiskers indicate minimum/maximum range. Differences between groups were evaluated by two-sided Mann–Whitney *U*-tests. *p*-values < 0.05 were used to denote a significant result. Plasma concentrations of IL-8, IL-6, IL-10, IL-15, IL-16, IL-17A, IFN-γ, and TNF in **b** maternal blood [$n = 8$ control, 11 SARS-CoV-2(+)] and **c** cord blood [$n = 7$ control, 9 SARS-CoV-2(+)]. Data are shown as boxplots where midlines indicate medians, boxes indicate interquartile range and whiskers indicate minimum/maximum range. Gray dotted lines indicate the lower limit of detection. Differences in cytokine concentrations between groups were evaluated by linear mixed-effects models with adjustment for covariates. Scatter plots of two principal components (PC1 and PC3 or PC1 and PC2) from plasma cytokine concentrations in the **d** maternal blood [$n = 8$ control, 11 SARS-CoV-2(+)] and **e** cord blood [$n = 7$ control, 9 SARS-CoV-2(+)]. The association between principal components (PC1, PC2, and PC3 jointly for maternal blood and PC2 for cord blood) and SARS-CoV-2 (+) status was assessed by logistic regression. Blue dots indicate control women, light red dots indicate SARS-CoV-2 (+) women, and dark red dots indicate women with severe COVID-19. Significant differences are based on *p* < 0.05.

infected with SARS-CoV-2 had increased concentrations of IL-8 (2-FC) compared to those born to control mothers (Fig. 1c, Supplementary Table 2); such an inflammatory change was not driven by the severe COVID-19 cases. However, no other significant differences in the concentrations of maternal and cord blood cytokines were observed between women infected with SARS-CoV-2 and control mothers (Fig. 1b, c, Supplementary Figs. 1 and 2, Supplementary Table 2). The maternal and cord blood cytokine responses, as captured by the first three principal components, were associated with the SARS-CoV-2 infection status ($p = 0.034$ for maternal blood and $p = 0.032$ for cord blood Fig. 1d, e). These results show that a cytokine response is observed in both the maternal and neonatal circulation upon maternal infection with SARS-CoV-2.

**Pregnant women with SARS-CoV-2 infection, but not their neonates, undergo a T-cell reduction in the circulation.** Previous studies have shown that patients with moderate or severe COVID-19 display alterations in their cellular immune responses in the peripheral circulation[29–31]. Therefore, we investigated whether pregnant women with SARS-CoV-2 infection and their neonates had changes in their cellular immune repertoire, using immunophenotyping (Fig. 2a, Supplementary Figs. 3a and 4a). Immunophenotyping included the identification of general leukocyte subpopulations as well as monocyte, neutrophil, T-cell, and B-cell subsets. Neutrophil and monocyte function has also been implicated in the pathogenesis of SARS-CoV-2 infection[31–33]; therefore, reactive oxygen species (ROS) production by neutrophils and monocytes was also determined in maternal and cord blood (Supplementary Fig. 5a). No statistical differences were observed in the mother or the neonate in the total number of general leukocyte subpopulations or in the monocyte, neutrophil, activated T-cell, and B-cell subsets (Supplementary Figs. 3b–f and 4b–f). Although neutrophils and monocytes produced ROS when stimulated, no differences were found between SARS-CoV-2 cases and controls in the maternal blood (Supplementary Fig. 5a, b). Mild differences were observed in the cord blood (Supplementary Fig. 5a, c). Nonetheless, pregnant women with SARS-CoV-2 infection had reduced T-cell numbers, but their neonates did not display such a decline (Fig. 2b). Heatmap and principal component analysis (PCA) representations of the immunophenotyping of the maternal blood showed that SARS-CoV-2 infection mildly altered T-cell subsets (Fig. 2c, d). Specifically, pregnant women infected with SARS-CoV-2 had reduced numbers of CD4$^+$ T cells, including $T_{CM}$ and Th1-like cells, as well as CD8$^+$ T cells, including $T_{CM}$, $T_{EM}$, and Tc17-like cells (Fig. 3a, b). Such changes were not solely driven by the severe COVID-19 cases. Neonates born to women with SARS-CoV-2 infection did not display changes in the T-cell subsets that were affected in mothers (Fig. 3c). These data showed that pregnant women infected with SARS-CoV-2 undergo a reduction in T-cell subsets, including pro-inflammatory Th1- and Tc17-like cells, which is not translated to the neonatal T-cell repertoire.

**Single-cell RNA sequencing reveals perturbed maternal and fetal immune responses at the maternal–fetal interface of women with SARS-CoV-2 infection.** Next, we investigated whether SARS-CoV-2 infection in the mother could alter cellular immune responses in the placenta, the organ that serves as the lungs, gut, kidneys, and liver of the fetus[34,35]. We performed scRNA-seq of the placental tissues including the basal plate (BP) (placental villi (PV) and BP, PVBP) and the chorioamniotic membranes (CAM) from pregnant women with SARS-CoV-2 infection and controls, using established methods. Consistent with our previous studies[21,36], multiple cell clusters were identified in

the placental tissues including lymphoid and myeloid immune cells, trophoblast cell types, stromal cells, and endometrial/decidual cells as well as endothelial cells (Fig. 4a). Differences in abundance among cell type clusters were observed between placental compartments as well as between tissues from women with SARS-CoV-2 infection and those from controls (Fig. 4b, c). Further analysis revealed that the majority of the differentially expressed genes (DEGs, Supplementary Data 1) between SARS-CoV-2-positive cases and controls belong to immune cells from the CAM, namely maternal T cells (89 DEGs) and macrophages (12 DEGs) (Fig. 4d, e). Decidual and lymphatic endothelial decidual (LED) cells of maternal origin displayed 12 and 11 DEGs, respectively, between SARS-CoV-2 cases and controls. Notably, fetal stromal cells from the CAM were also largely affected by SARS-CoV-2 infection (59 DEGs, Fig. 4d, e). However, other fetal cell types (e.g., trophoblasts and T cells) in the CAM and PVBP were minimally altered by the presence of SARS-CoV-2 infection in the mother (Fig. 4d, e).

The effects of SARS-CoV-2 on gene expression in maternal T cells from the CAM and PVBP were compared to those from peripheral T cells from hospitalized COVID-19 patients[37], which we will refer to as the reference database hereafter. Maternal T-cell gene expression changes resulting from SARS-CoV-2 infection in the CAM were positively correlated with those in the reference database (T cells from patients with COVID-19) (Spearman's $\rho = 0.27$, $p = 0.004$; Fig. 5a), suggesting a significant degree of shared DEGs. Yet, maternal T-cell gene expression induced by SARS-CoV-2 in the CAM was also distinct, since 53 of the 89 identified DEGs were not found in the reference database. By contrast, maternal T-cell gene expression dysregulation in the PVBP was not correlated with that from the reference database (Spearman's $\rho = 0.009$, $p = 0.62$; Fig. 5a). Gene Ontology (GO) analysis revealed that the shared DEGs between maternal T cells in the CAM and the reference T-cell data included translational termination and elongation as well as mitochondrial translational termination and elongation (Supplementary Fig. 6).

Although most of the DEGs were detected in the maternal T cells in the CAM, maternal macrophages and other cell types such as maternal monocytes, maternal decidual cells and LED, fetal stromal cells and trophoblast cell types also contributed to the differential gene expression observed between SARS-CoV-2 cases and controls (Fig. 5b). The top upregulated and downregulated genes in maternal T cell and Macrophage-2, as well as fetal Stromal-1, are also displayed in Fig. 5c, showing that changes in gene expression were not always homogeneous across all the cells. Gene set enrichment analysis of the DEGs from the most affected cell clusters (maternal T cell, Decidual, and Macrophage-2 and fetal Stromal-1), using the Kyoto Encyclopedia of Genes and Genomes (KEGG) pathways, revealed that each cell type specifically contributed to the inflammatory response. For example, the TNF signaling pathway was enriched in Decidual transcripts, cytokine–cytokine receptor interaction was enriched in maternal Macrophage-2, coronavirus disease (COVID-19) was enriched in fetal Stromal-1, and degenerative diseases were associated with maternal T cells (Fig. 5d). Furthermore, GO and Reactome pathway analysis of the combined DEGs from all cell types in response to SARS-CoV-2 infection included interferon signaling, TNF signaling pathway, antigen presentation, and other cellular processes associated with viral responses (Supplementary Fig. 7). Lastly, STRING enrichment analysis of all DEGs showed that the interferon signaling pathway was enriched in the placental tissues of women with SARS-CoV-2 infection (Supplementary Fig. 8a).

Taken together, these data show that placentas from women with SARS-CoV-2 display alterations in their immune repertoire,

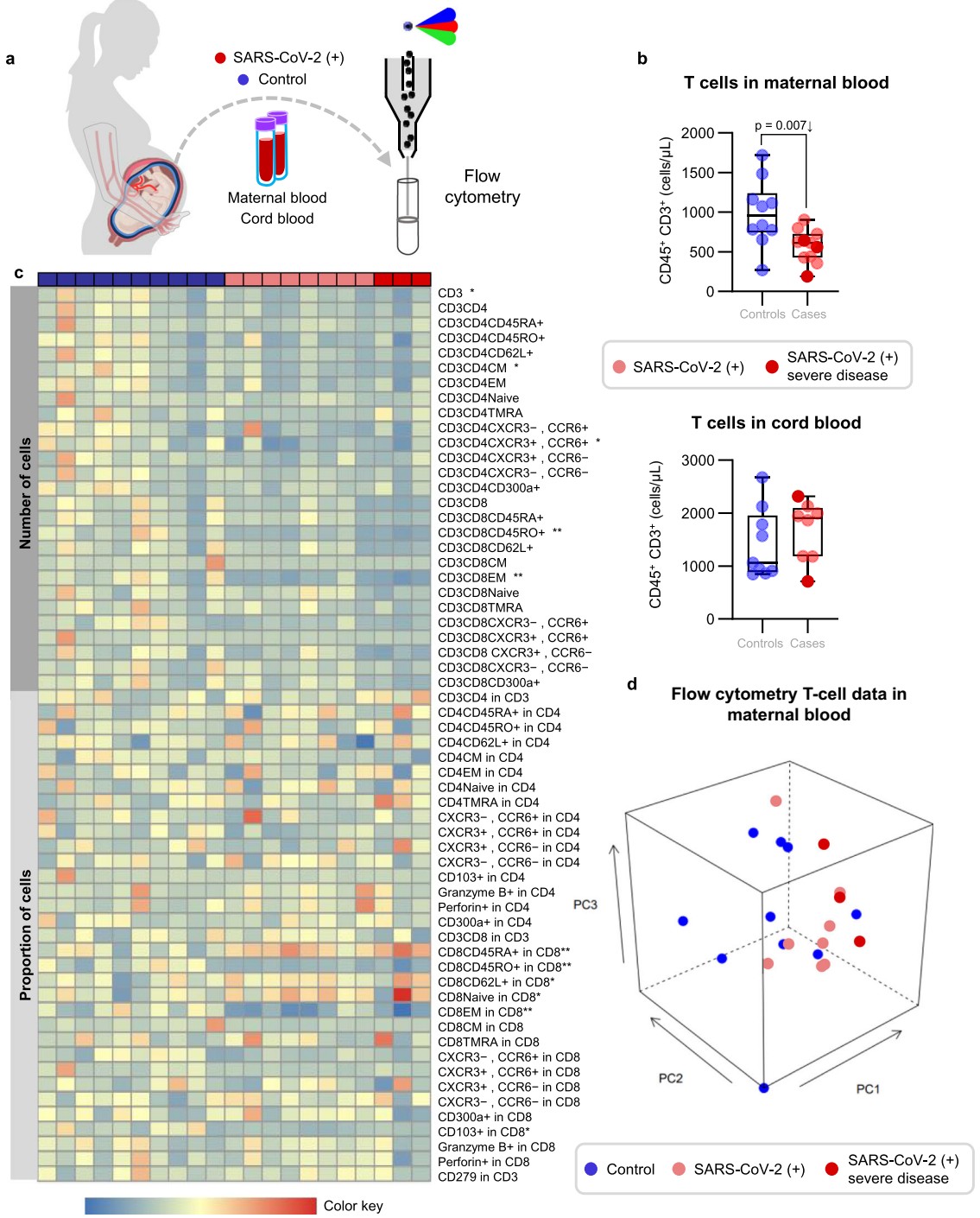

**Fig. 2 Immunophenotyping of T cells in the maternal and cord blood of women with SARS-CoV-2 infection and their neonates. a** Maternal blood and cord blood were collected for immunophenotyping. **b** Numbers of T cells in the peripheral blood [$n = 10$ control, 11 SARS-CoV-2 (+)] and cord blood [$n = 9$ control, 8 SARS-CoV-2 (+)]. Data are shown as boxplots where midlines indicate medians, boxes indicate interquartile range and whiskers indicate minimum/maximum range. Differences between groups were evaluated by Mann–Whitney $U$-tests. $P$ values < 0.05 were used to denote a significant result. Blue dots indicate control women, light red dots indicate SARS-CoV-2 (+) women, and dark red dots indicate women with severe COVID-19. **c** Heatmap showing the abundance ($z$-scores) of T-cell subsets in the maternal blood from SARS-CoV-2 (+) or control women [$n = 10$ control, 11 SARS-CoV-2 (+)], where cell numbers and proportions are shown. Differences between groups were assessed using two-sample $t$-tests. $P$ values were adjusted for multiple comparisons using the false-discovery rate (FDR) method to obtain $q$ values. *$q < 0.1$; **$q < 0.05$. Red and blue indicate increased and decreased abundance, respectively. **d** Three-dimensional scatter plot showing the distribution of flow cytometry data. Blue dots indicate control women, light red dots indicate SARS-CoV-2 (+) women, and dark red dots indicate women with severe COVID-19 [$n = 10$ control, 11 SARS-CoV-2(+)] based on principal component (PC)1–PC3.

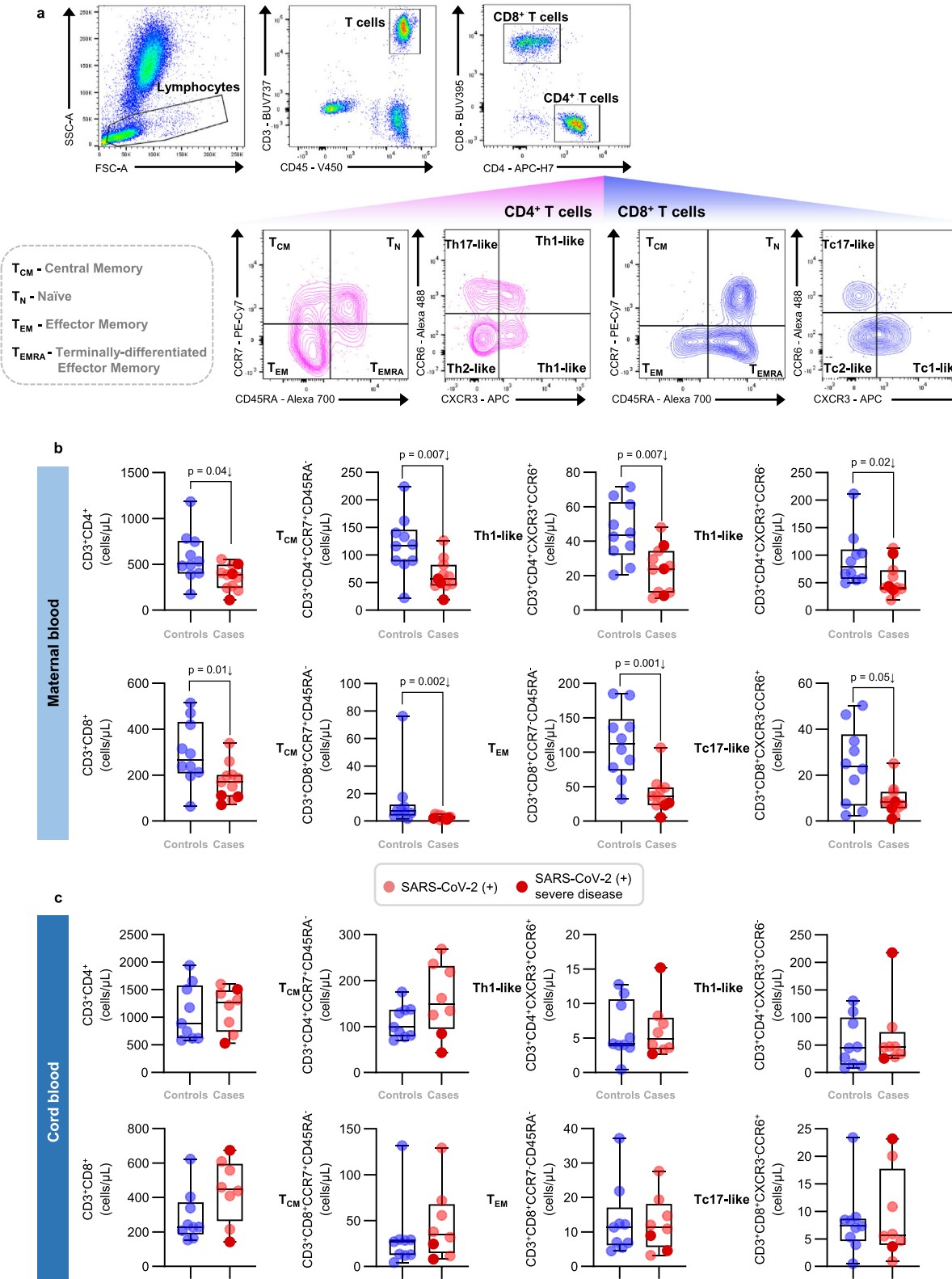

mainly in maternal T cells and macrophages infiltrating the CAM as well as their fetal stromal cells. Yet, the effect of SARS-CoV-2 on fetal T cells is minimal in our largely asymptomatic population.

**Blood RNA sequencing reveals shared and distinct immune responses to SARS-CoV-2 infection.** To further decipher the

effects of SARS-CoV-2 infection on the maternal and neonatal immune system, we performed RNA-sequencing of maternal blood and cord blood from cases and controls (Fig. 6a). The transcriptomes of maternal and cord blood were correlated (Spearman's $\rho = 0.24$; $p < 0.001$); therefore, some transcripts were shared between these two compartments (Fig. 6b). Yet, specific non-shared genes were modulated in the maternal

**Fig. 3 T-cell subsets in the maternal and cord blood of women with SARS-CoV-2 infection and their neonates. a** Representative gating strategy used to identify CD4$^+$ and CD8$^+$ T cells, and their respective subsets, within the total T-cell population (CD45$^+$CD3$^+$ cells) in the maternal blood and cord blood from SARS-CoV-2 (+) or control women. **b** Numbers of CD4$^+$ T cells, CD4$^+$ T$_{CM}$, CXCR3$^+$CCR6$^+$ Th1-like cells, and CXCR3$^+$CCR6$^-$ Th1-like cells (upper row); and numbers of CD8$^+$ T cells, CD8$^+$ T$_{CM}$, CD8$^+$ T$_{EM}$, and Tc17-like cells (lower row) in the maternal blood [$n = 10$ control, 11 SARS-CoV-2 (+)]. **c** Numbers of CD4$^+$ T cells, CD4$^+$ T$_{CM}$, CXCR3$^+$CCR6$^+$ Th1-like cells, and CXCR3$^+$CCR6$^-$ Th1-like cells (upper row); and numbers of CD8$^+$ T cells, CD8$^+$ T$_{CM}$, CD8$^+$ T$_{EM}$, and Tc17-like cells (lower row) in the cord blood [$n = 9$ control, 8 SARS-CoV-2 (+)]. Data are shown as boxplots where midlines indicate medians, boxes indicate interquartile range and whiskers indicate minimum/maximum range. Blue dots indicate control women, light red dots indicate SARS-CoV-2 (+) women, and dark red dots indicate women with severe COVID-19. Differences between groups were evaluated by two-sided Mann–Whitney $U$-tests, where $p < 0.05$ is considered significant.

blood and cord blood (Fig. 6b). Specifically, SARS-CoV-2 infection was associated with the dysregulation of 425 transcripts in maternal blood: 165 upregulated and 260 down-regulated (Fig. 6c and Supplementary Data 2). The biological processes enriched in the upregulated DEGs in maternal blood included humoral responses such as complement activation, adaptive immune responses, and immunoglobulin-mediated immune response, whereas those enriched in downregulated DEGs included phagocytosis and extracellular matrix organization (Fig. 6d). KEGG pathway analysis of DEGs in the maternal blood revealed enrichment for protein digestion and absorption pathways (Supplementary Data 3). Similarly, SARS-CoV-2 infection caused the dysregulation of 425 transcripts in the cord blood: 131 upregulated and 294 downregulated (Fig. 6c and Supplementary Data 4). The biological processes enriched in the upregulated DEGs in cord blood included defense responses to fungus and bacterium, and no significant biological processes were enriched in the downregulated DEGs (Fig. 6e). KEGG pathway analysis of DEGs in the cord blood did not show significant enrichment (Supplementary Data 5). Interaction analysis revealed that SARS-CoV-2 infection induced significantly different responses in the maternal blood compared to the cord blood for 34 genes (Supplementary Data 6). These genes were enriched for cellular and humoral biological processes such as phagocytosis and complement activation in the maternal blood compared to the cord blood (Supplementary Fig. 9).

To integrate the maternal and neonatal immune responses with those observed in the placental tissues, a correlation analysis was performed between bulk RNA-seq blood data and scRNA-seq placental data (Fig. 6f). Notably, CAM maternally derived scRNA-seq signatures of T cell, Macrophage-2, and Monocyte were correlated with the maternal blood transcriptome, and the PVBP fetally derived scRNA-seq T cell signature was negatively correlated with the cord blood transcriptome (Fig. 6g).

Collectively, these results indicate that SARS-CoV-2 infection differentially impacts the transcriptome of the mother and the neonate and that such changes are partly shared with those in the placental tissues. These findings also suggest that, although SARS-CoV-2 infection does not trigger fetal hematopoietic immune responses in the placenta as evidenced by our scRNA-seq data, it affects the neonatal immune system.

**SARS-CoV-2 RNA and proteins are not detected in the placentas of infected women.** SARS-CoV-2 induced unique maternal immune and fetal stromal responses in the extra-placental membranes; therefore, we explored whether this virus was present in the placenta. First, using a scRNA-seq approach, Viral-Track[38], we explored whether viral sequences were detected in the scRNA-seq data of CAM and PVBP from women with SARS-CoV-2 infection. SARS-CoV-2 viral sequences were detected in positive controls (bronchoalveolar lavage of patients infected with SARS-CoV-2[38]) but not in the placental tissues

from women with SARS-CoV-2 infection (Supplementary Fig. 8b, c).

Subsequently, we investigated the presence of viral RNA in the CAM, BP, and PV using RT-qPCR for the N1 and N2 viral genes (Supplementary Fig. 10a). SARS-CoV-2 N1 and N2 proteins were not detected in any of the placental samples from women with SARS-CoV-2 infection or healthy controls (Supplementary Fig. 10b). Yet, in the spike-in positive control, N1 and N2 RNA was detected in the CAM, BP, and PV (Supplementary Fig. 10b). A sensitivity assay revealed that 10 is the minimum confident copy number of viral particles detectable in the PV using RT-qPCR (Supplementary Fig. 10c).

Next, we determined whether the spike and nucleocapsid proteins were detected in the placental tissues of women with SARS-CoV-2 infection using immunohistochemistry (Fig. 7a). Several histological slides from the CAM, BP, and PV were included in our evaluation, including negative and spike-in positive controls (Supplementary Table 3). Both SARS-CoV-2 spike and nucleocapsid proteins were identified in the spike-in positive controls in the CAM, PV, and BP (Fig. 7b). A few of the placentas from asymptomatic women with SARS-CoV-2 infection displayed a putative positive signal for the spike and nucleocapsid proteins (Fig. 7c); yet, in all other cases, the placental tissues were negative for the SARS-CoV-2 proteins (Fig. 7d). As expected, spike and nucleocapsid SARS-CoV-2 proteins were not detected in the placental tissues of control women (Fig. 7e). To verify the detection of SARS-CoV-2 in the placental tissues, RNA was isolated from the same FFPE tissue sections where the putative positive signals were observed as well as those from some cases with negative signals, and RT-qPCR for the N1 and N2 viral genes was performed. FFPE tissue sections from the placental tissues of control women and spike-in positive controls were also included. None of the placentas from women with SARS-CoV-2 infection or controls had detectable levels of N1 and N2 RNA viral genes; yet, the spike-in positive controls were detected (Fig. 7f).

Collectively, these RT-qPCR and histologic data show that SARS-CoV-2 is not detected in the placental tissues, including the CAM, of women infected with SARS-CoV-2.

**SARS-CoV-2 infection during pregnancy does not compromise the sterility of the placenta.** The traditional view is that the placenta is a sterile organ that is first colonized by vaginal microbes during delivery.[39,40] However, the sterility of the placenta could be compromised by microorganisms invading from the lower genital tract (i.e., ascending infection) and those present in the maternal circulation (i.e., hematogenous infection).[41,42] Therefore, we evaluated whether infection with the SARS-CoV-2 virus, which can be detected in vaginal fluid[15] or the peripheral circulation,[43] could compromise the sterility of the placenta. Specifically, we used 16S rRNA gene qPCR and sequencing to characterize the bacterial DNA load and profiles of the amnion–chorion interface of the extraplacental CAM, the amnion–chorion interface of the placental disc, and the placental villous tree (Fig. 8a). As expected, mode of delivery was the principal factor affecting the bacterial DNA load (Supplementary

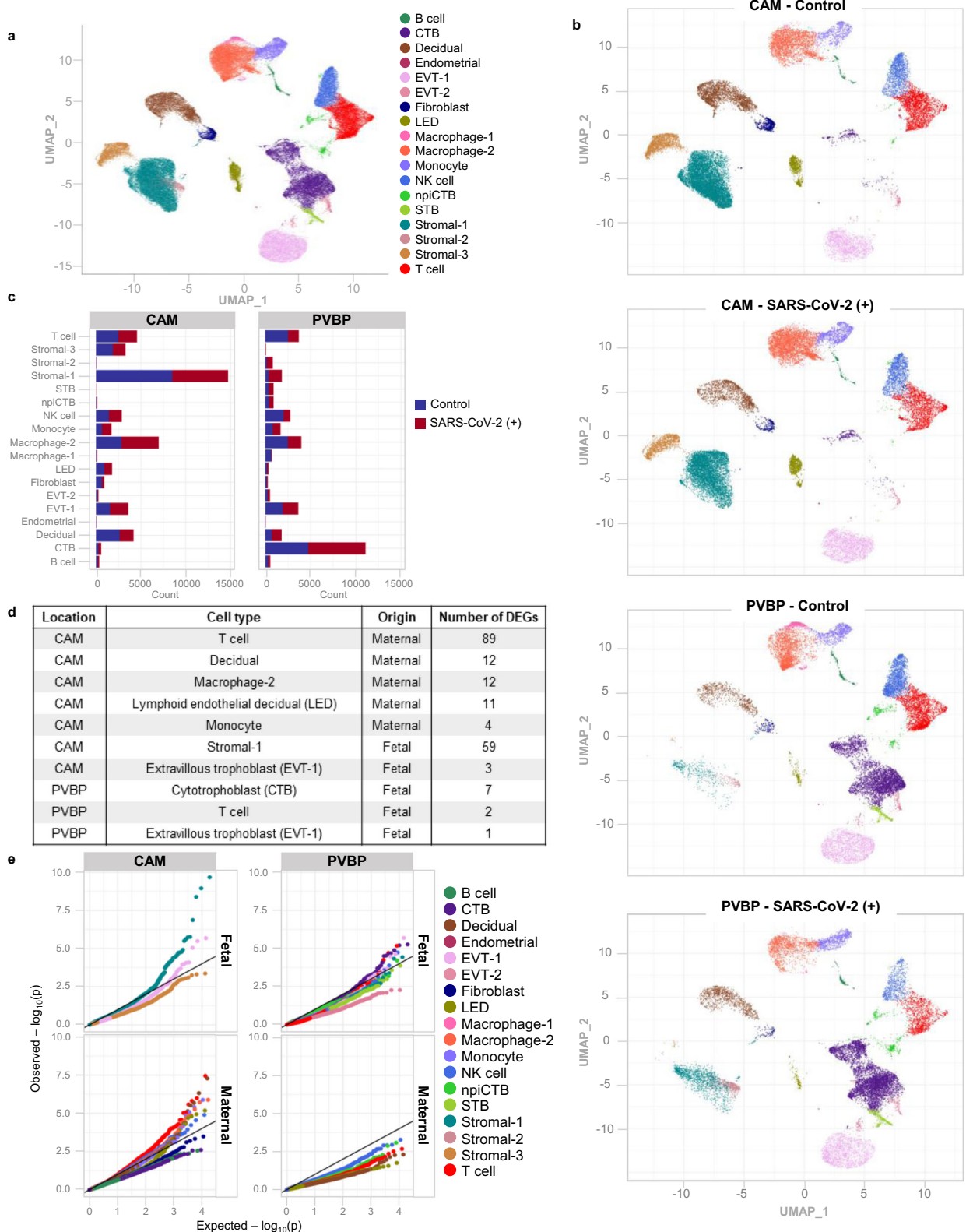

Table 4) and profile. Very few samples (4/15) from cesarean deliveries had a bacterial DNA load exceeding that of technical controls for background DNA contamination (i.e., blank DNA extraction kits), yet almost all of the samples (29/30) from vaginal deliveries did (Fig. 8b). Furthermore, whereas the bacterial DNA profiles of samples from cesarean deliveries were similar to those

of technical controls, those from vaginal deliveries were distinct, being dominated by DNA signals from *Lactobacillus* and *Ureaplasma*, similar to the vaginal swab positive controls (Fig. 8c). Among the samples obtained from vaginal deliveries, there was no difference in the bacterial DNA profiles based on maternal SARS-CoV-2 infection status (Fig. 8d). These findings show that,

**Fig. 4 Single-cell transcriptomics of the placental tissues from women with SARS-CoV-2 infection. a** Uniform Manifold Approximation and Projection (UMAP) plot showing the combined cell type classifications of the chorioamniotic membranes (CAM) and placental villi and basal plate (PVBP) from SARS-CoV-2 (+) ($n = 9$) or control women ($n = 10$), where each dot represents a single cell. Abbreviations used are CTB cytotrophoblast, EVT extravillous trophoblast, LED lymphoid endothelial decidual cell, npiCTB non-proliferative interstitial cytotrophoblast, STB syncytiotrophoblast. **b** UMAP plots showing cell populations separated based on placental compartment (CAM and PVBP) from SARS-CoV-2 (+) or control women. **c** Bar plots showing the numbers of each cell type in the CAM and PVBP of SARS-CoV-2 (+) or control women. **d** Numbers of differentially expressed genes (DEGs) associated with SARS-CoV-2 infection from the CAM and PVBP with false discovery rate (FDR) adjusted $p < 0.1$. **e** Quantile–quantile (Q–Q) plots showing the differential expression of all tested genes in each cell type of maternal or fetal origin from the CAM and PVBP samples. Deviation above the 1:1 line (solid black line) indicates enrichment.

although the mode of delivery alters the bacterial DNA loads and profiles of the placental tissues, we did not find evidence that the same is true for maternal SARS-CoV-2 infection.

## Discussion

This study provides evidence that, in a largely asymptomatic population, SARS-CoV-2 infection in pregnancy is primarily associated with maternal inflammatory responses in the circulation and at the maternal–fetal interface. First, we showed that pregnant women with SARS-CoV-2 infection had elevated levels of IgM and IgG in the peripheral circulation, whereas only IgG was detectable in the cord blood of their neonates, suggesting that acute fetal infection did not occur. This finding is consistent with several reports showing that IgM is undetected in the cord blood of neonates born to women with SARS-CoV-2 infection[26,44]. However, few studies have demonstrated that both IgM and IgG are detectable in a small fraction of neonates born to women diagnosed with COVID-19[15,23,27]. The increased levels of IgG in the cord blood are explained by the fact that this immunoglobulin crosses the placenta via the neonatal Fc receptor (FcRn), which is highly expressed in the syncytiotrophoblast layer[45]. Yet, it has been recently reported that, in the third trimester, the mechanisms whereby SARS-CoV-2-specific IgG1 crosses the placenta are compromised due to altered glycosylation profiles[46]. By contrast, IgM cannot cross the placenta given its large molecular weight, thus the detection of this immunoglobulin in the cord blood represents an acute fetal response in the clinical setting[47]. Therefore, the absence of detectable IgM in the cord blood suggests that vertical transmission *in utero* of SARS-CoV-2 was unlikely to occur in our study population.

In the current study, we report that pregnant women mount a mild systemic inflammatory response to SARS-CoV-2 characterized by increased concentrations of IL-8, IL-10, and IL-15, which is consistent with observations in non-pregnant individuals with SARS-CoV-2 infection[29]. Interestingly, we found that both SARS-CoV-2-infected mothers and their neonates had increased levels of IL-8 in their circulation. Interleukin-8 is a canonical pro-inflammatory cytokine whose primary function is neutrophil recruitment to sites of injury[48]. Relevant to this investigation, recent studies have proposed that IL-8 can serve as a biomarker for the prediction of disease severity and survival prognosis of patients infected with SARS-CoV-2[49]. Taken together, these data indicate that SARS-CoV-2 infection not only causes a maternal cytokine response but also induces neonatal inflammation, which can lead to long-term morbidities. Yet, the mechanisms whereby maternal SARS-CoV-2 infection elicits cytokine responses in the fetal/neonatal circulation require further investigation. A possibility is that the increased concentrations of IL-8 in the cord blood are explained by the transfer of maternal cytokines through the placental tissues, a process that has been documented in vivo for other pro-inflammatory cytokines[50].

Interestingly, neonates born to SARS-CoV-2-infected women displayed dysregulated immune and non-immune processes including the activation of neutrophils, which was previously

reported in children with COVID-19[51]. Such activation may be attributed to the elevated concentrations of IL-8 in the cord blood since this cytokine can activate neutrophils[52], suggesting that neonates born to women with SARS-CoV-2 infection display a mild neutrophil response. Even a seemingly mild immune response should not be overlooked as viral infections during pregnancy (influenza, Zika virus, etc.) have historically resulted in adverse long-term outcomes[53,54]. Pertinent to this concept, elevated concentrations of IL-8 in newborns are associated with the development of encephalopathies[55].

A hallmark of SARS-CoV-2 infection is lymphopenia, which is primarily reflected in the T-cell compartment[56–58], but not consistently observed for B cells[59]. Specifically, patients with symptomatic COVID-19 displayed reduced numbers of CD4+ and CD8+ T-cell subsets including naïve, central memory, and effector memory cells[31,58,60]. Lymphopenia is also correlated with COVID-19 disease severity, as critically ill patients showed the lowest numbers of total lymphocytes, including T cells, compared to asymptomatic individuals[61]. Yet, asymptomatic or mildly ill pregnant women seem to have slightly reduced lymphocyte numbers when compared to healthy controls[62]. Indeed, a recent single-center study showed that 80% of pregnant women with mild or asymptomatic SARS-CoV-2 infection displayed lymphopenia[63]. Consistently, we found that pregnant women with SARS-CoV-2 infection had reduced T-cell numbers compared to healthy controls, which included specific subsets such as CD4+ $T_{CM}$, Th1-like, CD8+ $T_{EM}$, and Tc17-like cells. Both Th1 and Tc17 cells participate in orchestrating pro-inflammatory responses in health and disease[64,65]. During pregnancy, these T-cell subsets are implicated in the establishment and maintenance of maternal–fetal tolerance, which plays a central role in pregnancy success[66–68]. Hence, these results indicate that SARS-CoV-2 infection alters specific pro-inflammatory T-cell subsets in the maternal circulation, which may compromise the mechanisms of maternal–fetal tolerance.

Concurrent with the cellular and humoral immune changes occurring in the periphery of pregnant women with SARS-CoV-2 infection, maternal T-cell responses in the CAM were also altered, as revealed by our scRNA-seq data. Maternal T cells reside at the maternal–fetal interface and their abundance changes as gestation progresses[68]. This T-cell compartment comprises multiple subsets, including effector/activated T cells, regulatory T cells, and exhausted T cells[68]. In addition, these adaptive immune cells can participate in the processes of labor by releasing inflammatory mediators such as TNF, IL-1β, and MMP-9[69]. The importance of T cells in the process of labor is underscored by observations showing that their single-cell signatures can be detected in the maternal circulation, providing a non-invasive approach to monitor pregnancy and its complications[36]. Consistent with these findings, herein we demonstrated that the single-cell signature of maternal T cells in the CAM from SARS-CoV-2-infected pregnant women resembled that of peripheral T cells from non-pregnant infected patients (obtained from a previously reported dataset[37]) and, more importantly, correlated with the cellular transcriptome of women infected with SARS-CoV-2. These results suggest that both systemic and local

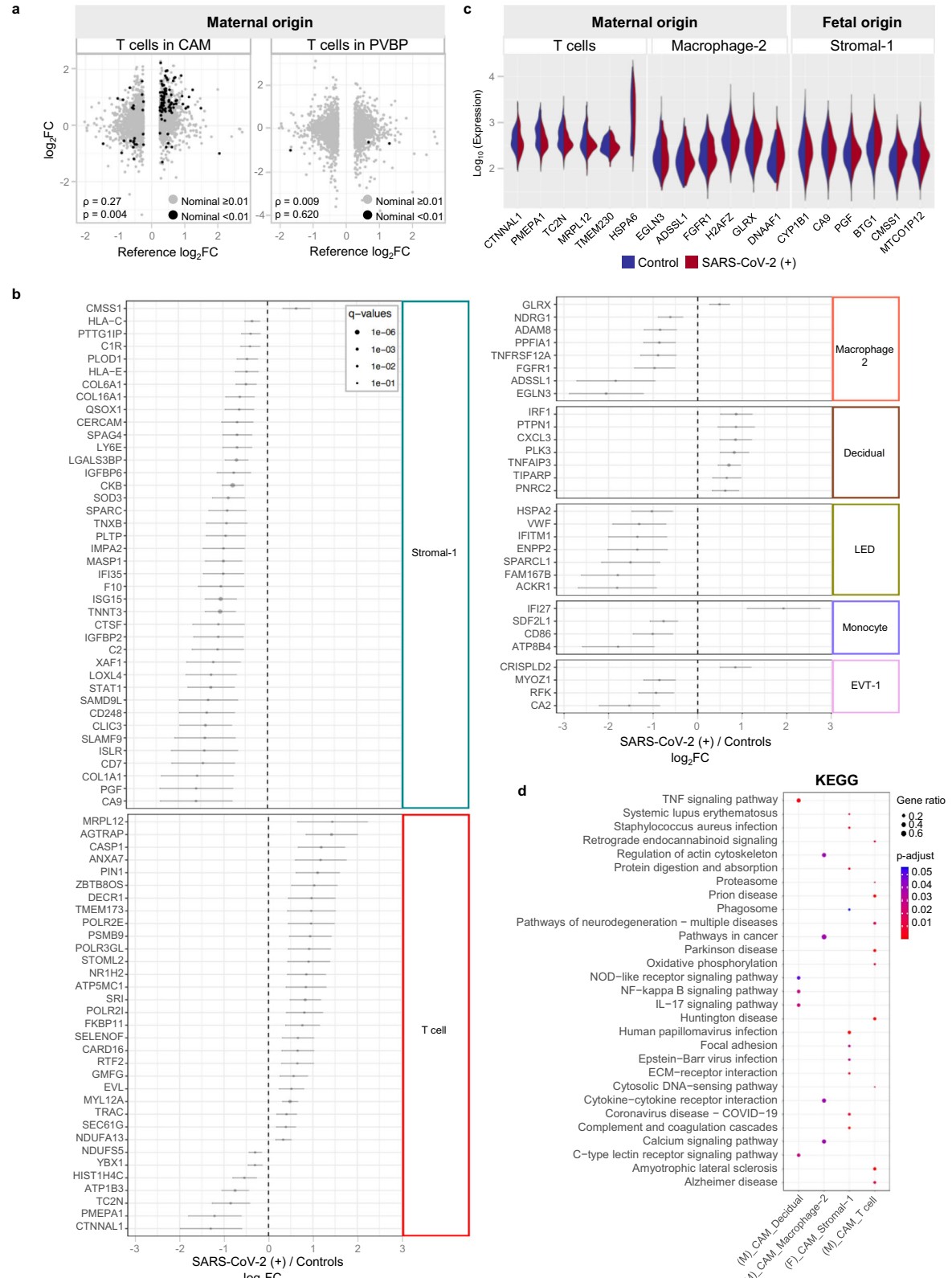

T-cell responses are altered by SARS-CoV-2; yet, pregnancy also promotes stereotypical cellular responses. Interestingly, maternal T cells from the CAM displayed enrichment of GO terms related to mitochondrial gene expression and translation, a process that has been implicated in T-cell functions including cytokine production[70].

Therefore, SARS-CoV-2 may enhance maternal T-cell function at the maternal–fetal interface.

SARS-CoV-2 infection also had effects on maternal macrophages in the CAM. In this compartment, macrophages can display pro- and anti-inflammatory functions that have been associated with

**Fig. 5 Single-cell characterization of major cell clusters in the chorioamniotic membranes (CAM) and placental villi and basal plate (PVBP) of women with SARS-CoV-2 infection. a** Scatter plots showing the effects of SARS-CoV-2 on gene expression [$\log_2$ fold change (FC)] in maternal T cells from the CAM and PVBP compared to a previously reported dataset (Meckiff et al., 2020). Black dots represent genes with nominal $p < 0.01$ in this study, which were used to calculate Spearman's correlation. **b** Forest plots showing the $\log_2$(FC) of differentially expressed genes (DEGs) associated with SARS-CoV-2 infection in Stromal-1, T cell, Macrophage-2, decidual, lymphoid endothelial decidual cell (LED), Monocyte, and extravillous trophoblast (EVT-1) cell types in the CAM and PVBP of SARS-CoV-2 (+) ($n = 9$) or control women ($n = 10$). DEGs shown are significant after false discovery rate (FDR) adjustment ($q < 0.1$). **c** Violin plots showing the distribution of single-cell gene expression levels for the top six DEGs in the maternal T cell, maternal Macrophage-2, and fetal Stromal-1 cell types in the CAM comparing SARS-CoV-2 (+) and control women. **d** Kyoto Encyclopedia of Genes and Genomes (KEGG) pathways enriched for DEGs in the maternal (M) cell types (decidual, T cells, and Macrophage-2) and fetal (F) Stromal-1 cell type from the CAM based on the over-representation analysis. A one-sided Fisher's exact test was used. KEGG pathways with $q < 0.05$ were selected.

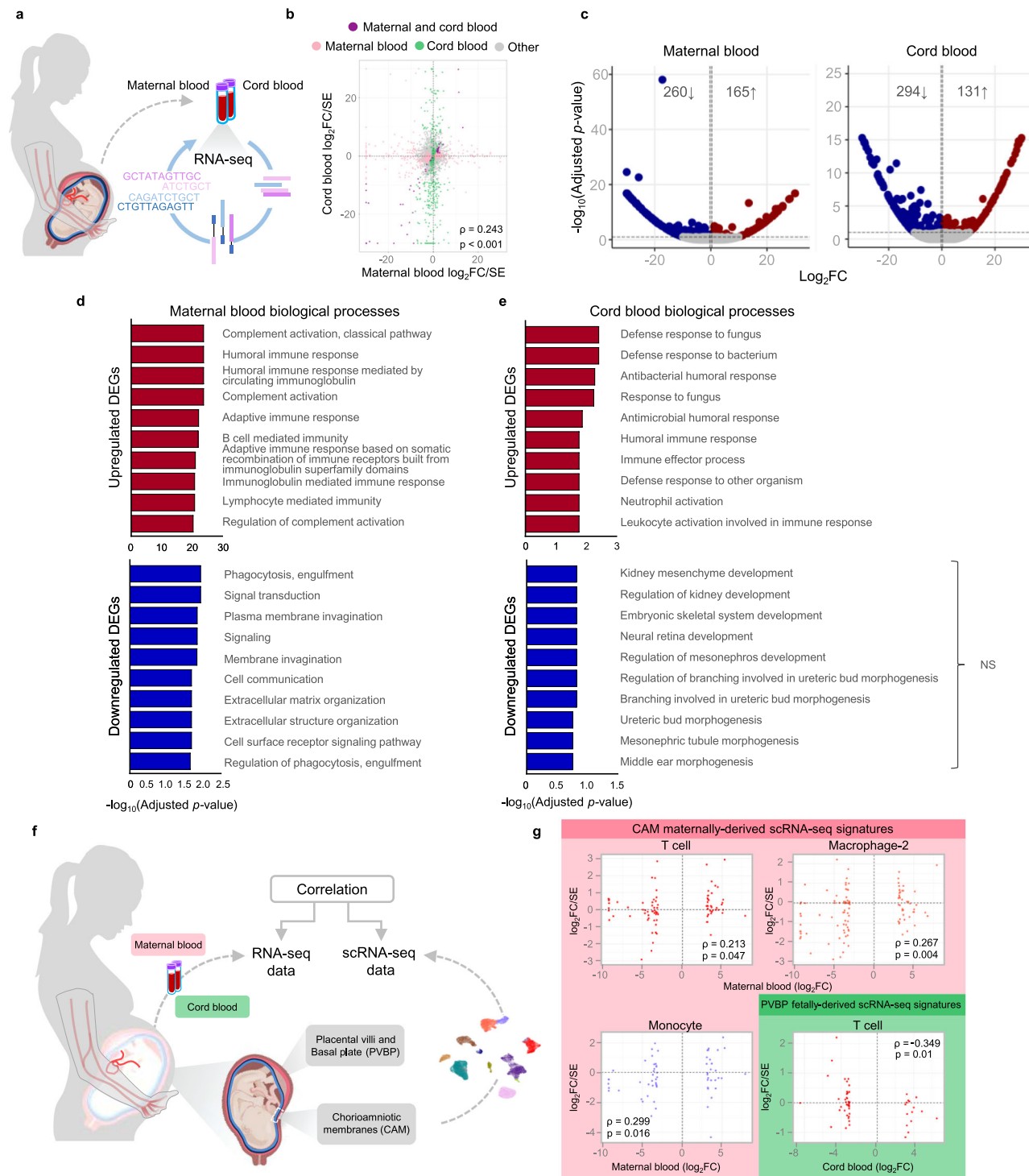

**Fig. 6 Bulk transcriptomics in the maternal and cord blood of women with SARS-CoV-2 infection and their neonates. a** Maternal blood [$n = 10$ control, 11 SARS-CoV-2 (+)] and cord blood [$n = 8$ control, 9 SARS-CoV-2 (+)] were collected for RNA sequencing (RNA-seq). **b** Scatter plot showing the $\log_2$ fold changes (FC) identified between SARS-CoV-2 (+) and control women that are similar or different between the maternal blood (x-axis) and cord blood (y-axis). A two-sided Spearman's correlation test was used. Each dot represents a gene tested (in pink DEG for the maternal blood only, in green DEG for the cord blood only, in purple DEG for both compartments and same direction, gray if otherwise not significant). **c** Volcano plots showing the adjusted $p$-values (-$\log_{10}$ thereof) on the y-axis and $\log_2$ fold change with SARS-CoV-2 infection on the x-axis in the maternal and cord blood. The number of upregulated and downregulated genes were determined based on fold change > 1.25 and adjusted $p$ value < 0.1. Bar plots showing the top 10 over-represented biological processes from the Gene Ontology (GO) database based on upregulated (top panel; red bars) and downregulated (bottom panel; blue bars) DEGs associated with SARS-CoV-2 infection with adjusted $p$ value < 0.05 in the **d** maternal blood and **e** cord blood. NS not significant. The $p$-values were computed based on one-sided hypergeometric distribution and adjusted by false-discovery rate. **f** Comparative analyses were performed between bulk transcriptomic data of the maternal blood and cord blood and scRNA-seq data from the placental tissues using DEG from the bulk analysis ($q < 0.1$). **g** Scatter plots (pink boxes) showing cell-type-specific Spearman correlations between bulk RNA-seq data from the maternal blood and scRNA-seq data from the chorioamniotic membranes (CAM) for the T cell, Macrophage-2, and Monocyte populations of maternal origin. A two-sided Spearman's correlation test was used. Scatter plot (green box) showing the cell type-specific Spearman correlation between bulk RNA-seq data from the cord blood and scRNA-seq data from the placental villi and basal plate (PVBP) for the T cell population of fetal origin.

pregnancy complications (e.g., preterm birth[71]) and maintenance[72], respectively. In the current study, the processes and pathways enriched in these tissue-resident innate immune cells included cytokine-cytokine receptor interaction, highlighting the role of macrophages in the inflammatory response against SARS-CoV-2 infection[73]. Notably, single-cell signatures of maternal macrophages positively correlated with the whole-blood transcriptome of women with SARS-CoV-2 infection, which was enriched for monocyte/macrophage-driven processes such as the classical pathway of complement activation. Indeed, such a humoral innate immune response is implicated in the pathogenesis of COVID-19[74]. Therefore, in the choriodecidual space, maternal macrophages participate in both tissue homeostasis and host defense as demonstrated during SARS-CoV-2 infection herein.

Importantly, we report that, although SARS-CoV-2 infection during pregnancy was neither associated with alterations in the neonatal (cord blood) T-cell repertoire nor fetal hematopoietic immune responses in the placenta, the transcriptome of fetal stromal cells in the CAM was profoundly impacted. Stromal cells play a central role in immunity by serving as physical barriers against microbes as well as actively participating in antigen presentation and T-cell responses.[75] However, the functions of fetal stromal cells are largely unknown, thus our findings represent the first insight into the involvement of these cells in viral infection. Yet, fetal innate and adaptive immune cells in the placenta were minimally affected by maternal SARS-CoV-2 infection. These observations are in tandem with the absence of SARS-CoV-2 transcripts/proteins in the placental tissues as well as undetectable IgM in the cord blood. Our results are in agreement with numerous reports showing that SARS-CoV-2 is undetected in the placenta[23], amniotic fluid[76], and neonates[23,26,76]. Yet, SARS-CoV-2 has been reported in the placentas of severe COVID-19 patients[14,25], indicating that this virus can on rare occasions reach and infect this organ. Therefore, the absence of SARS-CoV-2 in the CAM, PV, and BP of our mostly asymptomatic study population is in accordance with the known scarcity of placental infection[77].

Traditionally, the placenta is considered a sterile organ[39,40]. Indeed, recent research has reiterated the sterile womb hypothesis using placentas from women who delivered via cesarean section at term without labor[78,79] as well as studies in mice[80] and non-human primates[81]. Here, we evaluated the possibility that maternal SARS-CoV-2 infection compromises the sterility of the placenta by facilitating the invasion of bacteria or the transfer of bacterial DNA from maternal compartments. Consistent with our previous studies[78], the placentas of women who delivered via cesarean section did not consistently harbor a microbiome. Women who delivered vaginally displayed placental bacterial signatures similar to those from the lower genital tract; yet,

maternal SARS-CoV-2 infection did not modify such signatures. Hence, SARS-CoV-2 infection does not affect placental sterility in mostly asymptomatic women who delivered a term neonate.

It is worth mentioning that the cross-sectional design of this study has inherent limitations. For example, the diagnosis of SARS-CoV-2 infection in pregnant women was performed at the time of admission to the Labor and Delivery unit, and although our cases had elevated systemic concentrations of IgM (a sign of acute infection), our findings do not allow us to infer a timeline of severity or disease progression. Furthermore, our conclusions should be interpreted with caution since the number of severe cases is limited due to the rarity of COVID-19 during pregnancy[82]. Nonetheless, our study represents the first molecular characterization of the immune effects of SARS-CoV-2 during pregnancy in the mother, placenta, and offspring.

In summary, we have shown that SARS-CoV-2 infection during pregnancy primarily induces unique inflammatory responses at the maternal–fetal interface, which are largely governed by maternal T cells and fetal stromal cells. SARS-CoV-2 infection during pregnancy was also associated with humoral and cellular immune responses in the maternal blood, as well as with a mild cytokine response in the neonatal circulation (i.e., umbilical cord blood) without compromising the T-cell repertoire or initiating IgM responses. Importantly, SARS-CoV-2 was not detected in the placentas of infected women, nor was the sterility of the placenta compromised by this virus. This study provides insight into the maternal–fetal immune responses triggered by SARS-CoV-2 and further emphasizes the rarity of placental infection.

## Methods

**Human participants, clinical specimens, and definitions**. Human maternal peripheral blood, umbilical cord blood, and placental tissues were obtained at the Perinatology Research Branch, an intramural program of the *Eunice Kennedy Shriver* National Institute of Child Health and Human In Development (NICHD), National Institutes of Health, U.S. Department of Health and Human Services, Wayne State University (Detroit, MI, USA), and the Detroit Medical Center (DMC) (Detroit, MI, USA). The collection and use of human materials for research purposes were approved by the Institutional Review Boards of Wayne State University School of Medicine and the Detroit Medical Center. All participating women provided written informed consent prior to sample collection. The study groups were divided into pregnant women who had a positive RT-PCR test for SARS-CoV-2 (nasopharyngeal test provided by the Detroit Medical Center) and healthy gestational age-matched controls. The demographic and clinical characteristics of the study groups are shown in Supplementary Table 1. The maternal peripheral blood was collected at admission, prior to the administration of any medication, and the umbilical cord blood and placental tissues were collected immediately after delivery.

Gestational age was established based on the last menstrual period and confirmed by ultrasound examination. Labor was defined as the presence of regular uterine contractions with a frequency of ≥2 times every 10 min and cervical ripening. Term delivery was defined as birth ≥37 weeks of gestation. Preeclampsia was defined as new-onset hypertension that developed ≥20 weeks of gestation and

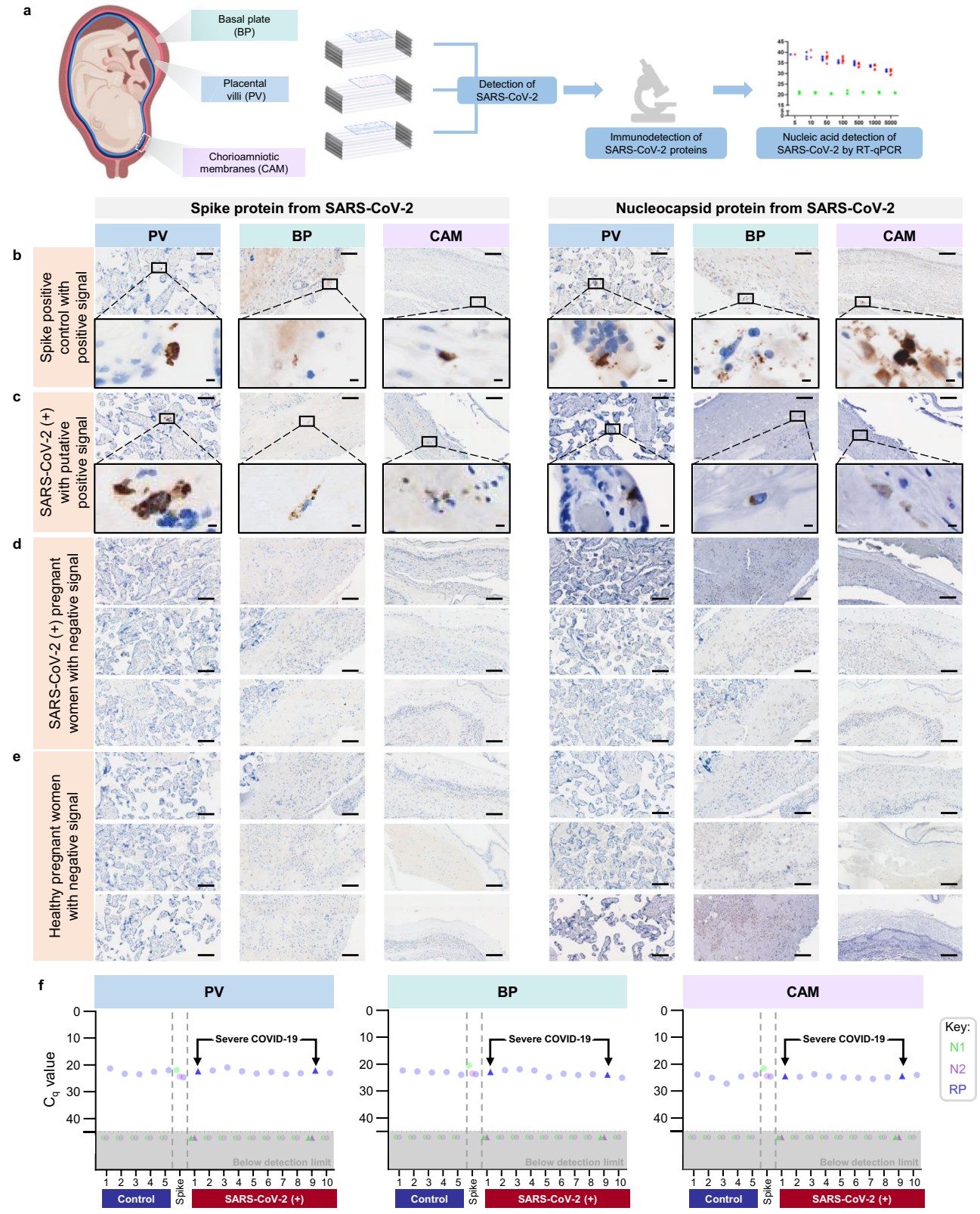

proteinuria. Other clinical and demographic characteristics were obtained by review of medical records.

**Placental histopathological examination.** Placentas were examined histologically by perinatal pathologists according to standardized DMC protocols. Briefly, three to nine sections of the placenta were examined, and at least one full-thickness section was taken from the center of the placenta; others were taken randomly from the placental disc. Acute and chronic inflammatory lesions of the placenta (maternal inflammatory response and fetal inflammatory response), as well as other placental lesions, were diagnosed according to established criteria[83], as shown in Supplementary Table 1.

**Immunoassays**

*Immunoglobulin (Ig) M and G determination in the maternal blood and umbilical cord blood.* Maternal peripheral blood and umbilical cord blood were collected into

**Fig. 7 Immunohistological and molecular detection of SARS-CoV-2 proteins/RNA in placentas of women with SARS-CoV-2 infection. a** Schematic representation showing various sampling locations in the placental villi (PV), basal plate (BP), and chorioamniotic membranes (CAM) of SARS-CoV-2 (+) or control women that were evaluated for SARS-CoV-2 protein detection by immunohistochemistry, followed by RNA viral detection using RT-qPCR [$n = 5$ control, 10 SARS-CoV-2(+)] (see Supplementary Table 3). **b** Images showing a positive signal for SARS-CoV-2 spike (left panel) and nucleocapsid (right panel) proteins in the PV, BP, and CAM of spike-in positive control. **c** Images showing a putative positive signal for SARS-CoV-2 spike (left panel) and nucleocapsid (right panel) proteins in the PV, BP, and CAM of a SARS-CoV-2 (+) woman. Images are representative of two independent experiments. **d** Three representative images showing a negative signal for SARS-CoV-2 spike (left panel) and nucleocapsid (right panel) proteins in the PV, BP, and CAM of SARS-CoV-2 (+) women ($n = 10$). **e** Three representative images showing a negative signal for SARS-CoV-2 spike (left panel) and nucleocapsid (right panel) proteins in the PV, BP, and CAM of control women ($n = 5$). Brown staining indicates a putative positive signal. All images were taken at 200× magnification. Scale bars represent 100 μm (or 5 μm for digital zoom-in images). **f** SARS-CoV-2 viral RNA detection by RT-qPCR in the PV, BP, and CAM from histological slides of SARS-CoV-2 (+) ($n = 10$) and control women ($n = 5$). N1 (green dot/triangle) and N2 (purple dot/triangle) denote two SARS-CoV-2 nucleocapsid (N) genes, and RP (blue dot/triangle) denotes the human RNase P gene, which serves as a positive internal PCR control. Triangles indicate women with severe COVID-19. Spike-in positive controls were also included. Undetermined quantification cycle ($C_q$) values are represented below the detection limit (gray area).

tubes without an anticoagulant, and the tubes were stored at room temperature for 30–60 min prior to centrifugation for 10 min at 1600×$g$ and 4 °C. After centrifugation, the serum was collected and stored at −80 °C. The serum concentrations of SARS-CoV-2 IgM and IgG were determined using the human anti-SARS-CoV-2 IgM and human anti-SARS-CoV-2 IgG ELISA kits (LifeSpan BioSciences, Inc., Seattle, WA, USA), according to the manufacturer's instructions. Plates were read using the SpectraMax iD5 (Molecular Devices, San Jose, CA, USA) and analyte concentrations were calculated with the SoftMax Pro 7 (Molecular Devices). The sensitivities of the assays were 0.469 ng/mL (human anti-SARS-CoV-2 IgM) and 2.344 ng/mL (human anti-SARS-CoV-2 IgG).

*Determination of cytokine and chemokine concentrations in the maternal blood and umbilical cord blood.* Maternal peripheral blood and umbilical cord blood were collected into tubes with an anticoagulant (EDTA or citrate), which were centrifuged for 10 min at 1600×$g$ and 4 °C. Upon centrifugation, the plasma was collected and stored at −80 °C prior to cytokine/chemokine determination. The V-PLEX Pro-Inflammatory Panel 1 (human) and Cytokine Panel 1 (human) immunoassays (Meso Scale Discovery, Rockville, MD, USA) were used to measure the concentrations of IFN-γ, IL-1β, IL-2, IL-4, IL-6, IL-8, IL-10, IL-12p70, IL-13, and TNF (Pro-inflammatory Panel 1) or GM-CSF, IL-1α, IL-5, IL-7, IL-12/IL-23p40, IL-15, IL-16, IL-17A, TNF-β, and VEGF-A (Cytokine Panel 1) in the maternal and cord blood plasma, according to the manufacturer's instructions. Plates were read using the MESO QuickPlex SQ 120 (Meso Scale Discovery) and analyte concentrations were calculated with the Discovery Workbench 4.0 (Meso Scale Discovery). The sensitivities of the assays were 0.21–0.62 pg/mL (IFN-γ), 0.01–0.17 pg/mL (IL-1β), 0.01–0.29 pg/mL (IL-2), 0.01–0.03 pg/mL (IL-4), 0.05–0.09 pg/mL (IL-6), 0.03-0.14 pg/mL (IL-8), 0.02-0.08 pg/mL (IL-10), 0.02-0.89 pg/mL (IL-12p70), 0.03-0.73 pg/mL (IL-13), 0.01–0.13 pg/mL (TNF), 0.08–0.19 pg/mL (GM-CSF), 0.05–2.40 pg/mL (IL-1α), 0.04–0.46 pg/mL (IL-5), 0.08–0.17 pg/mL (IL-7), 0.25–0.42 pg/mL (IL-12/IL-23p40), 0.09–0.25 pg/mL (IL-15), 0.88–9.33 pg/mL (IL-16), 0.19–0.55 pg/mL (IL-17A), 0.04–0.17 pg/mL (TNF-β), 0.55–6.06 pg/mL (VEGF-A).

*Immunophenotyping of maternal and cord blood leukocytes.* Maternal peripheral blood and umbilical cord blood were collected into tubes containing EDTA. Fifty microlitres of whole blood were incubated with fluorochrome-conjugated anti-human mAbs (Supplementary Table 5) for 30 min at 4 °C in the dark. After incubation, erythrocytes were lysed using BD FACS lysing solution (BD Biosciences, San Jose, CA, USA). For intracellular staining, erythrocyte lysis was not performed and the cells were instead fixed and permeabilized using the BD Cytofix/Cytoperm kit (BD Biosciences) prior to staining with intracellular fluorochrome-conjugated anti-human mAbs (Supplementary Table 5). Finally, leukocytes were washed and resuspended in 0.5 mL of FACS staining buffer (BD Biosciences) and acquired using the BD LSRFortessa flow cytometer and FACSDiva 6.0 software. The absolute number of cells was determined using CountBright absolute counting beads (Thermo Fisher Scientific/Molecular Probes, Eugene, OR, USA). The analysis and figures were performed using the FlowJo software version 10 (FlowJo, Ashland, OR, USA). Immunophenotyping included the identification of general leukocyte populations (neutrophils, monocytes, T cells, B cells, and NK cells), monocyte subsets, neutrophil subsets, T-cell subsets, and B-cell subsets. Specifically, the numbers of effector memory T cells ($T_{EM}$; CD3+CD4+/CD8+CD45RA−CCR7−), naïve T cells ($T_N$; CD3+CD4+/CD8+CD45RA+CCR7+), central memory T cells ($T_{CM}$; CD3+CD4+/CD8+CD45RA−CCR7+), terminally differentiated effector memory T cells ($T_{EMRA}$; CD3+CD4+/CD8+CD45RA+CCR7−), Th1/Tc1-like T cells (CD3+CD4+/CD8+CXCR3+CCR6+/CCR6-), Th2/Tc2-like T cells (CD3+CD4+/CD8+CXCR3−CCR6−), and Th17/Tc17-like T cells (CD3+CD4+/CD8+CXCR3-CCR6+) in maternal and cord blood are presented in Fig. 3.

*ROS production by neutrophils and monocytes.* Fifty microlitres of maternal peripheral blood and cord blood were stimulated with 50 μL of ROS assay mix

containing 1:250 of ROS assay stain and ROS assay buffer [both from the ROS assay kit (eBioscience, San Diego, CA, USA)] and 1 μL of phorbol myristate acetate (PMA; 3 μg/mL) (Millipore Sigma, Burlington, MA, USA). The unstimulated group received 1:250 ROS assay mix and 1× phosphate-buffered saline (PBS) (Thermo Fisher Scientific/Gibco, Grand Island, NY, USA). The cells were incubated at 37 °C with 5% $CO_2$ for 60 min. Following incubation, erythrocytes were lysed using ammonium–chloride–potassium (ACK) lysing buffer (Lonza, Walkersville, MD, USA), and the resulting leukocytes were collected after centrifugation at 300×$g$ for 5 min. Next, leukocytes were resuspended in 0.5 mL of 1× PBS and acquired using the BD LSRFortessa flow cytometer and FACSDiva 6.0 software to measure ROS production by neutrophils and monocytes. The analysis and figures were performed using the FlowJo software version 10.

### Single-cell RNA sequencing

*Preparation of single-cell suspensions.* Single-cell suspensions were prepared from the BP, PV, and CAM, as previously described with modifications[36]. Digestion of placental tissues was performed using collagenase A (Sigma Aldrich, St. Louis, MO, USA) or the enzyme cocktail from the Umbilical Cord Dissociation Kit (Miltenyi Biotec, San Diego, CA, USA). Next, tissue suspensions were washed with 1× PBS and filtered through a cell strainer (Miltenyi Biotec). Cell pellets were collected after centrifugation at 300×$g$ for 10 min at 20 °C. Erythrocytes were lysed using ACK lysing buffer and the reaction was stopped by washing with 0.04% bovine serum albumin (Sigma Aldrich) in 1× PBS. Then, the cell pellets were collected after centrifugation at 300×$g$ for 10 min at 20 °C and resuspended in 1× PBS for cell counting using an automatic cell counter (Cellometer Auto 2000; Nexcelom Bioscience, Lawrence, MA, USA). Dead cells were removed from the cell suspensions using the Dead Cell Removal Kit (Miltenyi Biotec) to obtain a final cell viability of ≥80%.

*Single-cell library preparation using the 10× Genomics platform.* Viable cells were used for single-cell RNA-seq library preparation following the protocol for the 10× Genomics Chromium Single Cell 3′ Gene Expression Version 3 Kit (10× Genomics, Pleasanton, CA, USA). Briefly, cell suspensions were loaded into the Chromium Controller to generate gel beads in emulsion (GEM), each containing a single cell and a single Gel Bead with barcoded oligonucleotides. Reverse transcription of mRNA into complementary (c)DNA was performed using the Veriti 96-well Thermal Cycler (Thermo Fisher Scientific, Wilmington, DE, USA). The resulting cDNA was purified using Dynabeads MyOne SILANE (10× Genomics) and the SPRIselect Reagent (Beckman Coulter, Indianapolis, IN, USA). cDNA amplicons were optimized via enzymatic fragmentation, end-repair, and A-tailing followed by the incorporation of adapters and sample index by ligation. The sample index PCR product was amplified using the Veriti 96-well Thermal Cycler. The Agilent Bioanalyzer High Sensitivity Chip (Agilent Santa Clara, CA, USA) was used to analyze and quantify the final library construct. The Kapa DNA Quantification Kit for Illumina platforms (Kapa Biosystems, Wilmington, MA, USA) was used to quantify the DNA libraries, following the manufacturer's instructions.

*Sequencing.* 10× scRNA-seq libraries were sequenced on the Illumina NextSeq 500 in the Genomics Services Center (GSC) of the Center for Molecular Medicine and Genetics (Wayne State University School of Medicine, Detroit, MI, USA). The Illumina 75 Cycle Sequencing Kit (Illumina, San Diego, CA, USA) was used with 58 cycles for R2, 26 for R1, and 8 for I1.

*Genotyping.* DNA was extracted from maternal peripheral blood and umbilical cord blood/tissue using DNeasy Blood and Tissue Kit (Qiagen, Hilden, Germany), following manufacturer's instructions modified with the addition of 4 μl RNase A (100 mg/mL) (Qiagen) and incubation in 56 °C. Purified DNA samples were quantified using Qubit™ dsDNA HS Assay Kit (Invitrogen, Carlsbad, CA, USA). Two platforms were used for genotyping: (i) low-coverage (~0.4×) whole-genome

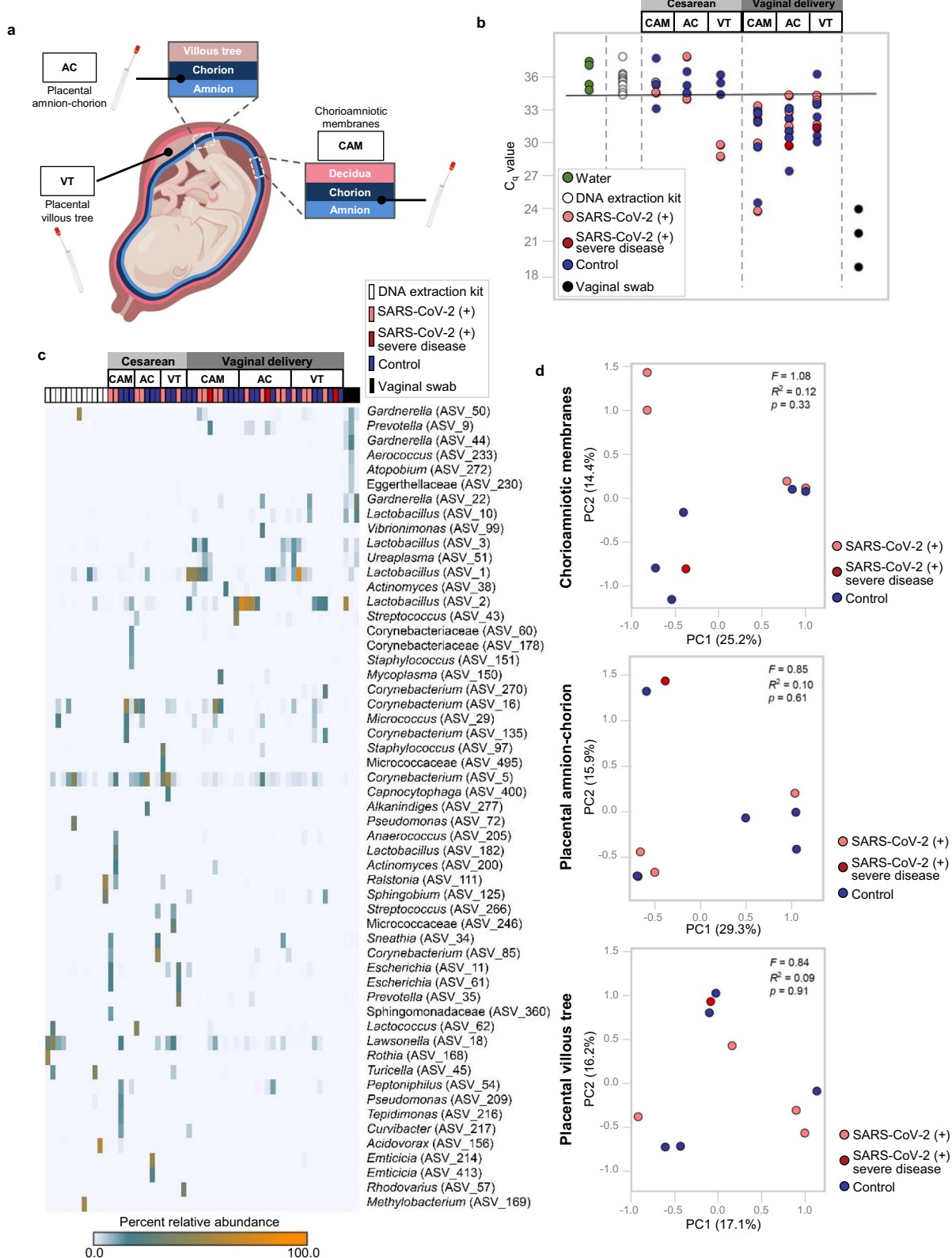

sequencing imputed to 37.5 M variants using the 1000 Genomes database (Gencove, New York, NY, USA); and (ii) Infinium Global Diversity Array-8 v1.0 Kit microarrays processed by the Advanced Genomics Core of University of Michigan (Ann Arbor, MI, USA). For the array platform, genotype information was converted to vcf format using "iaap-cli gencall" and "gtc_to_vcf.py" from Illumina, and imputation to 37.5 M variants using the 1000 Genomes haplotype references was done using the University of Michigan Imputation Server (https://imputationserver.sph.umich.edu/). The maternal/fetal relationship of the genotyped samples was ascertained using plink2

KING-robust kinship analysis[84]. The vcf files from the two platforms were then merged together and filtered for high-quality imputation and coverage for at least 10 scRNA-seq transcripts using bcftools.

*scRNA-seq data analysis*. Sequencing data were processed using Cell Ranger version 4.0.0 from 10× Genomics for de-multiplexing. The fastq files were then aligned using kallisto[85], and bustools[86] summarized the cell/gene transcript counts in a

**Fig. 8 Bacterial DNA profiles of the placental tissues from women with SARS-CoV-2 infection. a** Schematic representation of sampling locations from the chorioamniotic membranes (CAM), amnion–chorion interface of the placenta (AC), and within the placental villous tree (VT) from SARS-CoV-2 (+) women who delivered by cesarean section [$n = 3$ control, 2 SARS-CoV-2(+)] or vaginally [$n = 5$ control, 5 SARS-CoV-2(+)]. **b** Quantitative real-time PCR analyses illustrating the bacterial loads (i.e., 16S rRNA gene abundance) of the CAM, AC, and VT from SARS-CoV-2 (+) or control women (cesarean section or vaginal delivery). The solid line denotes the lowest cycle of quantification (i.e., highest bacterial load) for any blank DNA extraction kit negative control. Data from three human vaginal swabs are included for perspective. **c** Heatmap illustrating the relative abundances of prominent (>2% average relative abundance) amplicon sequence variants (ASVs) among the 16S rRNA gene profiles of the CAM, AC, and VT from SARS-CoV-2 (+) or control women (cesarean section or vaginal delivery). Data from blank DNA extraction kit negative controls and human vaginal swabs are included for perspective. **d** Principal coordinates analyses (PCoA) illustrating similarity in the 16S rRNA gene profiles of the CAM, AC, and VT obtained through vaginal delivery from SARS-CoV-2 (+) or control women. Statistical analysis was performed using permutational multivariate analysis of variance (PERMANOVA) through the "adonis" function. Blue dots indicate control women, light red dots indicate SARS-CoV-2 (+) women, and dark red dots indicate women with severe COVID-19.

matrix for each sample using the "lamanno" workflow for scRNA-seq. Each library was then processed using DIEM[87] to eliminate debris and empty droplets. In parallel, "cellranger counts" were also used to align the scRNA-seq reads using the STAR[88] aligner to produce the bam files necessary for demultiplexing the individual of origin, based on the genotype information using souporcell[89] and demuxlet[90]. We removed any droplet/GEM barcode that was assigned to doublet or ambiguous cells in demuxlet or souporcell, and only those cells that could be assigned a pregnancy case and maternal/fetal origin were kept. All count data matrices were then normalized and combined using the "NormalizeData," "Find-VariableFeatures," and "ScaleData" methods implemented in the Seurat package in R (Seurat version 3.1, R version 4.0.0)[91,92]. Next, the Seurat "RunPCA" function was applied to obtain the first 100 principal components, and the different libraries were integrated and harmonized using the Harmony package in R version 1.0[93]. The top 30 harmony components were then processed using the Seurat "runU-MAP" function to embed and visualize the cells in a two-dimensional map via the Uniform Manifold Approximation and Projection for Dimension Reduction (UMAP) algorithm[94,95]. To label the cells, the SingleR[96] package in R version 1.3.8 was used to assign a cell-type identity based on our previously labeled data as a reference panel (as performed in ref. [36]). Cell type abbreviations used are STB, syncytiotrophoblast; EVT, extravillous trophoblast; CTB, cytotrophoblast; npiCTB, non-proliferative interstitial cytotrophoblast; LED, lymphoid endothelial decidual cell; and NK, natural killer cell. Genes utilized to distinguish Macrophage-1, Macrophage-2, Stromal-1, Stromal-2, and Stromal-3 cell types are provided in Supplementary Data 7.

*Differential gene expression for scRNA-seq data.* To identify differentially expressed genes, we created a pseudo-bulk aggregate of all the cells of the same cell type/location/origin. For each combination, we only used samples with more than 20 cells. The negative binomial model implemented in the DESeq2 R package version 1.28.1[97] was used to calculate the log$_2$ FC between SARS-CoV-2 (+) and healthy pregnant women. A term for each library was added to the DESeq2 model to correct for technical batch effects. We also evaluated the contribution of additional covariates, but their impact was minimal when compared to the model that adjusts for batch effects only. The p-values were adjusted using the Benjamini-Hochberg false-discovery rate method (FDR)[98], and the DEGs were selected based on an adjusted p-value < 0.1. qqplot was used to assess the distribution of the p-values and to identify which cell types and location combinations have higher enrichment for low p-values. Forest plots were used to visualize the DEGs (adjusted p-value ≤ 0.05), with each dot representing the log$_2$FC of the SARS-CoV-2 (+) group and the bars representing the 95% confidence interval. The genes with the highest log$_2$FC across T-cell, Macrophage-2, and Stromal-1 cell types were further illustrated using violin plots representing the single-cell gene expression data in log$_{10}$[transcripts per million].

*Comparison with previous scRNA-seq SARS-CoV-2 studies.* Single-cell RNA-seq data showing the effects of SARS-CoV-2 on peripheral T cells were obtained from a previous study[37]. The log$_2$FCs from this previous study were compared to those obtained here in maternal T cells from the PV and BP (PVBP) and the CAM. The comparison was visualized with scatter plots using the ggplot2 R package version 3.3.2 and Spearman's correlation analysis. Additionally, this previously generated set of SARS-CoV-2-associated genes in T cells was used to repeat the FDR p-value adjustment to reduce the burden of multiple testing in CAM-derived maternal T cells and provided a long list of genes. This list of genes was further analyzed with the clusterProfiler in R version 4.0.1 to perform over-representation analysis (ORA).

*GO and pathway enrichment analysis for scRNA-seq data.* The clusterProfiler in R version 4.0.1[99] was used to perform ORA based on the GO, KEGG (release 90.0+ / 05-29[100,101]), and Reactome databases. P-values were adjusted for multiple comparisons using the FDR method[98]. The functions "enrichPathway" and "enrichKEGG" from "clusterProfiler" were used to perform ORA separately for each list of genes obtained as differentially expressed for each cell type, placental compartment,

and maternal/fetal origin. Only results that were significant after correction were reported with q < 0.05 being considered statistically significant.

*STRING analysis.* The STRING database (https://string-db.org) was utilized to identify and visualize the enrichment of DEGs from the Reactome database, regardless of cell type, compartment, and origin. The STRING database integrates the known and predicted protein-protein associations from many organisms, including both direct (physical) and indirect (functional) interactions[102].

*Analysis of viral reads in scRNA-seq libraries.* The R-based computational pipeline Viral-Track was used to study viruses in raw scRNA-seq data (github.com/PierreBSC/Viral-Track)[38]. A combined index of both the host GRCH37(hg19) and viral reference genomes was constructed in Viral-Track. The viral genomes were downloaded from the Virusite database version 2020.3[103] that includes all published viruses, viroids, and satellites (NCBI RefSeq). Afterward, the STAR aligner was used to map reads to the indexed host and viral genome. Viral genomes were filtered based on read-map quality, nucleotide composition, sequence complexity, and genome coverage. Sequence complexity was calculated by computing the average nucleotide frequency and Shannon's entropy. Reads with a sequence entropy above 1.2, genome coverage greater than 5%, and longest contig longer than three times the mean read length are required for a viral segment to be considered present (default thresholds empirically defined by Viral-Track). As no viral reads were detected in our PVBP and CAM libraries, the correct implementation of the Viral-Track pipeline was validated by reanalyzing the data of bronchoalveolar lavage samples of patients with severe and mild SARS-CoV-2[38] and reproducing the detection of SARS-CoV-2 and human metapneumovirus.

## RNA sequencing of maternal blood and cord blood

*RNA isolation and sequencing.* Total RNA was isolated from the maternal blood and cord blood collected into PAXgene Blood RNA tubes using the PAXgene Blood RNA Kit (Qiagen), following the manufacturer's instructions. The purity, concentration, and integrity of the RNA samples were assessed using the Nano-Drop 1000 spectrophotometer (Thermo Scientific, Wilmington, Delaware, USA) and the Bioanalyzer 2100 (Agilent Technologies, Wilmington, Delaware, USA). The RNA-seq library was prepared by BGI Genomics (BGI ShenZhen, China) using the Oligo dT Stranded mRNA library preparation protocol (BGI, Hong Kong). Paired-end sequence reads (at least 50 million paired-end reads) of 150 bp length were generated using DNBseq (MGI-G400), and the raw data were provided by BGI.

*Bulk RNA sequencing data analysis.* Transcript abundance from RNA-seq reads was quantified with Salmon[104]. The differential expression of genes between groups was tested by fitting a negative binomial distribution model implemented in DESeq2. The model included maternal age, BMI, nulliparity, labor status, and delivery route as covariates. Genes were filtered out if not detected in at least two samples regardless of the infection status. Genes with a minimum FC of 1.25-fold and an adjusted p-value (q-value) of <0.1 were considered differentially expressed. The DEGs for each group comparison were used as input in iPathwayGuide (ADVAITA Bioinformatics, Ann Arbor, MI, USA)[105,106] to identify the significantly impacted biological processes and pathways. Volcano plots were used to display the evidence of differential expression for each comparison. The differences in SARS-CoV-2-associated gene expression changes between maternal and cord blood were tested using negative binomial models implemented in the DESeq2 package. These models included the main effects of disease status, sample type (maternal blood vs. cord blood), and their interactions. A minimum difference in FC of 1.25 and a FDR-adjusted p-value (q-value) < 0.1 was considered significant.

*Comparison between single-cell transcriptomics from gestational tissues and bulk transcriptomes from maternal and umbilical cord blood.* The log$_2$FC associated with SARS-CoV-2 in the maternal blood and cord blood was compared to those obtained from immune cell types in scRNA-seq of the placental tissues. For each

tissue and/or cell type, the FC was obtained from the DESeq2 model as described above by comparing SARS-CoV-2 (+) pregnancies to healthy controls. The FC were standardized by dividing by the standard error provided by the DESeq2 model, given the difference in the two technologies. Only immune-relevant cell types that had differentially expressed genes in the single-cell analysis were used for comparison: maternal and fetal T cell (PVBP and CAM), maternal Macrophage-2 (CAM), and maternal Monocyte (CAM). The comparisons of the effects of SARS-CoV-2 were based on the Spearman correlation and visualized with scatter plots using the ggplot2.

## Detection of SARS-CoV-2 RNA/proteins in the placenta

*Detection of SARS-CoV-2 RNA in the placenta.* Total RNA was isolated from the BP, PV, and CAM using QIAshredders and the RNeasy Mini Kit (both from Qiagen), according to the manufacturer's instructions. Positive and negative controls were SARS-Related Coronavirus 2 (SARS-CoV-2) External Run Control and Negative Control (both from ZeptoMetrix, Buffalo, NY, USA). Following the instructions from the CDC-2019 Novel Coronavirus (2019-nCoV) Real-Time RT-PCR Diagnostic Panel, cDNA was synthesized using TaqPath™ 1-Step RT-qPCR Master Mix, CG (Thermo Fisher Scientific/Applied Biosystems, Frederick, MD, USA) and primers from the 2019-nCoV RUO Kit (Integrated DNA Technologies, Newark, NJ, USA). Reactions were incubated at 25 °C for 2 min followed by 50 °C for 15 min. Initial denaturation was set for 2 min at 95 °C followed by 45 amplification cycles at 95 °C for 3 s and 55 °C for 30 s. A cycle of quantification ($C_q$ value) less than 45 indicates a positive result. Two positive PCR controls were used: 2019-nCoV_N (virus) and Hs_RPP30 (human) (both from Integrated DNA Technologies). Each PCR sample was run in duplicate.

RNA extractions were also performed using QIAamp Viral RNA Mini Kit (Qiagen) and results were comparable to those generated using the RNeasy Mini Kit.

*SARS-CoV-2 viral particle sensitivity assay.* For each experiment ($n = 3$), nine pieces of freshly collected PV were used. Eight of the tissue pieces were spiked with increasing numbers of viral particles [SARS-Related Coronavirus 2 (SARS-CoV-2) External Run Control] (0, 5, 10, 50, 100, 500, 1000, or 5000 particles/homogenate) and a piece of tissue was spiked with SARS-Related Coronavirus 2 (SARS-CoV-2) Negative Control prior to mechanical digestion. Total RNA was isolated using the RNeasy Mini Kit, according to the manufacturer's instructions. Detection of SARS-CoV-2 RNA was performed as described above.

*Detection of SARS-CoV-2 proteins in the placenta.* Five-μm-thick tissue sections of formalin-fixed, paraffin-embedded PV, BP, and the CAM were cut, mounted on SuperFrost™ Plus microscope slides (Erie Scientific LLC, Portsmouth, NH, USA), and subjected to immunohistochemistry using SARS-CoV/SARS-CoV-2 (COVID-19) spike antibody [1A9] [Catalog. No: GTX632604 (IHC-P application), dilution 1:100] (GeneTex, Irvine, CA, USA) and SARS-CoV-2 (COVID-19) nucleocapsid antibody [Catalog. No. GTX135357 (IHC-P application), dilution 1:100] (Gene-Tex). To serve as a positive control, placental tissues from pregnant women were spiked with SARS-CoV-2 (Isolate: USA/WA1/2020) (ZeptoMetrix) Culture Fluid (heat-inactivated). Spiked tissues were subjected to immunohistochemistry using SARS-CoV/SARS-CoV-2 (COVID-19) spike antibody [1A9] and SARS-CoV-2 (COVID-19) nucleocapsid antibody. Staining was performed using the Leica Bond-Max automatic staining system (Leica Microsystems, Wetzlar, Germany) with the Bond Polymer Refine Detection Kit (Leica Microsystems). The mouse isotype [Catalog. No: IS75061-2 (IHC-P application), no dilution needed] (Agilent) and rabbit isotype [Catalog. No: IS60061-2 (IHC-P application), no dilution needed] (Agilent) were used as negative controls. Tissue slides were then scanned using the Vectra Polaris Multispectral Imaging System (Akoya Biosciences, Marlborough, MA, USA) and images were analyzed using the Phenochart v1.0.8 image software (Akoya Biosciences). Supplementary Table 6 summarizes the number of slides included in this study.

*Detection of SARS-CoV-2 viral RNA in formalin-fixed paraffin-embedded placental tissues.* Formalin-fixed paraffin-embedded (FFPE) placental tissues that showed a positive signal for SARS-CoV-2 spike or nucleocapsid proteins as indicated by immunohistochemistry were used for detection of SARS-CoV-2 viral RNA. Total RNA was isolated from 6-14 sections of 10-μm-thick FFPE BP, PV, and the CAM using the PureLink™ FFPE Total RNA Isolation Kit (Invitrogen), according to the manufacturer's instructions. Total RNA was also isolated from spiked tissues as described above. Following the instructions from the CDC-2019 Novel Coronavirus (2019-nCoV) Real-Time RT-PCR Diagnostic Panel, cDNA was synthesized using TaqPath™ 1-Step RT-PCR Master Mix, CG, and primers from the 2019-nCoV RUO Kit. Reactions were incubated at 25 °C for 2 min followed by 50 °C for 15 min. Initial denaturation was set for 2 min at 95 °C followed by 45 amplification cycles at 95 °C for 3 s and 55 °C for 30 s. A cycle of quantification ($C_q$ value) less than 45 indicates a positive result. Two positive PCR controls were used: 2019-nCoV_N (virus) and Hs_RPP30 (human). Each PCR sample was run in duplicate.

## Molecular microbiology

*Sample collection.* Swabs (FLOQSwabs; COPAN, Murrieta, CA, USA) for molecular microbiology were collected from the CAM, the amnion–chorion interface of the placental disc, and the placental villous tree. These swabs were stored at −80 °C until DNA extractions were performed.

*DNA extraction.* All DNA extractions were completed within a biological safety cabinet using a DNeasy Powerlyzer Powersoil Kit (Qiagen, Germantown, MD, USA), with minor modifications to the manufacturer's instructions as previously described[80,81]. Personnel wore sterile surgical masks, gowns, and gloves during the procedure. Briefly, following UV treatment, 400 μL of Powerbead solution, 200 μL of phenol:chloroform:isoamyl alcohol (pH 7–8), and 60 μL of pre-heated solution C1 were added to the bead tubes. The swab samples were added to the tubes, incubated for 10 min, and then mechanically lysed for two rounds of 30 s each using a bead beater. Following a 1 min centrifugation and transferring of the supernatant to new tubes, 1.0 μL of PureLink™ RNase A (20 mg/mL) (Invitrogen), 100 μL of solution C2, and 100 μL of solution C3 were added. The tubes were incubated at 4 °C for 5 min and centrifuged for 1 min. After transferring the lysates to new tubes, 650 μL of solution C4 and 650 μL of 100% ethanol were added. Next, 635 μL of the lysate were loaded onto the filter columns and centrifuged for 1 min, discarding the flowthrough. This wash step was repeated three times to ensure all the lysates passed through the columns. Following the washes, 500 μL of solution C5 were added to the filter columns. After a 1 min centrifugation, the flowthrough was discarded and the tubes were centrifuged again for 2 min to dry the spin columns. The spin columns themselves were transferred to a clean 2.0 mL collection tube, and 60 μL of pre-heated solution C6 were added directly to the center of the spin columns. After a 5 min incubation at room temperature, the DNA was eluted via a 1 min centrifugation. Purified DNA was then transferred to clean 2.0 mL collection tubes and immediately stored at −20 °C. Twelve extractions of sterile FLOQSwabs were included as technical controls for potential background DNA contamination.

*16s rRNA gene quantitative real-time PCR.* Total bacterial DNA abundance within samples was measured via amplification of the V1–V2 region of the 16S rRNA gene according to the protocol of Dickson et al.[107] with minor modifications, as previously described[80,81]. These modifications included the use of a degenerative forward primer (27f-CM: 5′-AGA GTT TGA TCM TGG CTC AG-3′) and a degenerate probe containing locked nucleic acids (+) (BSR65/17: 5′-56FAM-TAA + YA + C ATG + CA + A GT + C GA-BHQ1-3′). Each 20 μL reaction contained 0.6 μM of 27f-CM primer, 0.6 μM of 357R primer (5′-CTG CTG CCT YCC GTA G-3′), 0.25 μM of BSR65/17 probe, 10.0 μL of 2× TaqMan Environmental Master Mix 2.0 (Invitrogen), and 3.0 μL of either purified DNA or nuclease-free water. The total bacterial DNA qPCR was performed using the following conditions: 95 °C for 10 min, followed by 40 cycles of 95 °C for 30 s, 50 °C for 30 s, and 72 °C for 30 s. Duplicate reactions were run for all samples.

Raw amplification data were normalized to the ROX passive reference dye and analyzed using the 7500 Software version 2.3 (Applied Biosystems, Foster City, CA, USA) with automatic threshold and baseline settings. The cycle of quantification (Cq) values were calculated for samples based on the mean number of cycles required for normalized fluorescence to exponentially increase.

*16S rRNA gene sequencing and processing.* Amplification and sequencing of the V4 region of the 16S rRNA gene were performed using the dual indexing sequencing strategy developed by Kozich et al.[108]. The forward primer was 515F: 5′-GTGCCA GCMGCCGCGGTAA-3′ and the reverse primer was 806R: 5′-GGACTACHVG GGTWTCTAAT-3′. Each PCR reaction contained 0.75 μM of each primer, 3.0 μL template DNA, 10.0 μL of 2× TaqMan Environmental Master Mix 2.0, and DNase-free water to produce a final volume of 20 μL. Reactions were performed using the following conditions: 95 °C for 10 min, followed by 40 cycles of 95 °C for 20 s, 55 °C for 15 s, and 72 °C for 5 min, with an additional elongation at 72 °C for 10 min. All PCR reactions were run in duplicate and products from duplicate reactions were pooled prior to purification and sequencing.

16S rRNA gene sequencing libraries were prepared according to Illumina's protocol for Preparing Libraries for Sequencing on the MiSeq (15039740 Rev. D) for 2 or 4 nM libraries. Sequencing was conducted using the Illumina MiSeq platform (V2 500 cycles, Illumina MS102-2003), according to the manufacturer's instructions with modifications found in[108]. All samples were quantified using the Qubit dsDNA HS assay and pooled in equimolar concentration prior to sequencing.

16S rRNA gene sequences were clustered into amplicon sequence variants (ASVs) defined by 100% sequence similarity using DADA2 version 1.12[109] in R version 3.6.1[110] according to the online MiSeq protocol (https://benjjneb.github.io/dada2/tutorial.html) with minor modifications, as previously described[81]. These modifications included allowing truncation lengths of 250 and 150 bases, and a maximum number of expected errors of 2 and 7 bases, for forward and reverse reads, respectively. To increase power for detecting rare variants, sample inference allowed for the pooling of samples. In addition, samples in the resulting sequence table were pooled prior to the removal of chimeric sequences. Sequences were then classified using the silva_nr_v132_train_set database with a minimum bootstrap value of 80%, and sequences that were derived from Archaea, chloroplast, or Eukaryota were removed.

The R package decontam version 1.6.0[111] was used to identify ASVs that were potential background DNA contaminants based on their pattern of occurrence in

biological vs. technical control samples using the "IsNotContaminant" function. An ASV was determined to be a contaminant, and was thus removed from the entire dataset, if it had a $p$ score $\geq 0.4$, had a higher mean relative abundance in technical controls than biological samples, and was present in more than one-third of technical control samples. Although one ASV, which was classified as *Lactobacillus*, met all the criteria for being defined as a contaminant, it was highly abundant in all three positive control vaginal samples and was therefore not removed from the dataset. Ultimately, a total of 148 ASVs determined to be contaminants were removed from the dataset prior to analysis. The vast majority of these ASVs were classified as *Staphylococcus* (138/148 ASVs; 93.2%).

*16S rRNA gene profile statistical analyses.* Prior to analyses, the dataset was randomly subsampled to 5426 sequences per sample. Heatmaps of the 16S rRNA gene profiles of samples, including all prominent ASVs (i.e., those ASVs with an average relative abundance $\geq 2\%$ for any placental site and/or mode of delivery combination) were generated using the open-source software program Morpheus (https://software.broadinstitute.org/morpheus). Differences in the structure of 16S rRNA gene profiles of samples were assessed using the Bray–Curtis dissimilarity index. Variation in the 16S rRNA gene profiles of the placental samples from different study groups was visualized through principal coordinates analyses using the R package vegan version 2.5-6[112]. Statistical evaluation of 16S rRNA gene profile differences between study groups was completed using permutational multivariate analysis of variance (PERMANOVA)[113] through the "adonis" function in the R package vegan version 2.5-6.

**Statistical analysis.** Statistical analyses were performed using SPSS v19.0 (IBM, Armonk, NY, USA) or the R package (as described above). For human demographic data, the group comparisons were performed using the Fisher's exact test for proportions and the Mann–Whitney $U$-test for non-normally distributed continuous variables. Immunoglobulin concentrations were compared using Mann–Whitney $U$-tests. Since cytokine data were obtained in duplicates, and one of the two measurements could have been below the detection limit, we used linear mixed-effects models to compare concentrations between groups while accounting for the number of measurements available in each sample. In these models, the fixed effects were the infection status, maternal age, BMI, and nulliparity, while a random effect was assigned to each patient. The significance of the group coefficient was assessed using the likelihood ratio test between a model with and without the infection status. Before fitting the models, cytokines with levels below the detection limit in both duplicates in a given sample were imputed with 99% of the smallest detected value across all samples. PCA of cytokine data after imputation of values below detection limit was performed using the R package PCAtools after separately normalizing the data from maternal and cord blood. A logistic regression model was used to test the association between the infection status and up to three principal components. Significant $p$-values were based on a likelihood ratio test. For the comparison of flow cytometry data between study groups, Mann–Whitney $U$-tests were performed. $p < 0.05$ was considered statistically significant. For heatmap representation of immunophenotyping results, flow cytometry data were transformed into $Z$-scores by subtracting the mean and dividing by the standard deviation, which were both calculated from the control group. The $Z$-scores were visualized as a heat map and compared between SARS-CoV-2 (+) and control groups using two-sample $t$-tests. $p$-values were adjusted for multiple comparisons using the FDR method to obtain $q$-values. A $q$-value $< 0.1$ was considered statistically significant. The principal components (PC) of the flow cytometry data were also determined, and PC1–PC3 were plotted on a 3D scatter plot. Single-cell RNA-seq and MiSeq data analyses were performed as described in their respective sections.

**Reporting summary.** Further information on research design is available in the Nature Research Reporting Summary linked to this article.

## Data availability
The majority of the data generated in this study are included in the manuscript or in the Supplementary Materials. The raw numbers for charts and graphs are available in the Source Data file whenever possible. The genotyping and single-cell RNA-seq data reported in this study were deposited in the NIH dbGAP repository (accession number phs001886.v3.p1). The raw MiSeq data reported in this study were deposited in the NCBI Sequence Read Archive (Bioproject ID: PRJNA701628). The bulk RNA-seq data from the maternal and cord blood were deposited in the Gene Expression Omnibus (https://www.ncbi.nlm.nih.gov/geo/; accession number GSE185557). All software and R packages used herein are detailed in the Materials and Methods.

## Code availability
Scripts detailing the single-cell analyses are available at https://github.com/piquelab/covid19placenta.

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

## Acknowledgements
We thank the physicians, nurses, and research assistants from the Center for Advanced Obstetrical Care and Research, Intrapartum Unit, Perinatology Research Branch Clinical Laboratory, and Perinatology Research Branch Perinatal Translational Science Laboratory for help with collecting samples, and Andrew Winters for his contribution to the microbiome analyses. This research was supported by the Perinatology Research Branch, Division of Obstetrics and Maternal–Fetal Medicine, Division of Intramural Research, *Eunice Kennedy Shriver* National Institute of Child Health and Human Development, National Institutes of Health, U.S. Department of Health and Human Services (NICHD/NIH/DHHS) under Contract No. HHSN275201300006C (R.R.). This research was also supported by the Wayne State University Perinatal Initiative in Maternal, Perinatal and Child Health (N.G.-L. and A.L.T.). R.R. has contributed to this work as part of his official duties as an employee of the United States Federal Government. The graphical representation of a fetus shown in Figs. 1, 2, and 6–8 was created by using a fully licensed version of BioRender.com.

## Author contributions
V.G.-F. performed experiments, analyzed data, and wrote the paper. N.G.-L. conceived, designed, and supervised the study, provided intellectual input, and wrote the paper. R.R. conceived and supervised the study, provided intellectual input, and wrote the paper. R.P.-R., Y.X., K.T., and A.T. designed the study, analyzed data, provided intellectual input, and wrote the paper. M.A.-H., D.M., A.P., G.B., J.G., and M.G. performed experiments or analyzed data, and drafted the paper. D.L., E.P., L.T., D.K., V.F., Y.L., K.M., and G.Z. and performed experiments or analyzed data. R.P., M.F., and C.-D.H. provided human samples used in the study and intellectual input. All authors revised and provided feedback for the final version of the paper.

## Competing interests
The authors declare no competing interests.
