## [Peer Review File · Nature Communications]

Reviewers' Comments:

Reviewer #1:

Remarks to the Author:

The manuscript by Garcia-Flores et al describes the immune responses of pregnant women that are SARS-CoV-2 infected at delivery and their placenta cell composition. The study is performed in seven pregnant women that test positive for SARS-CoV2 at delivery and 8 normal pregnant women. Blood was collected from the mothers and from the umbilical cord and shown analyzed for cytokines. It is unclear at this stage, whether those cytokines have similar probabilities to cross the placenta and whether the ones found in the cord blood are of embryonic origin. What appears clear though is that the T-cell lymphopenia observed in CoV patients is not present in the cord blood T cells. Neither are the local placental fetal immune responses. This is a carefully done study that shows that in mild cases of CoV 2 infection (there is a single severe case in this study) the placenta is rarely infected, there is maternal immune cell activation in the placenta but limited fetal immune reactions.

The authors have certainly data from neonatal T and B cells found in the placenta. It would be interesting to show and discuss the transcriptional profile of these cells compared to those from the mother and from control newborns.

Reviewer #2:

Remarks to the Author:

This is an interesting and broad study by Garcia-Flores and colleagues ("Maternal-Fetal Immune Responses in Pregnant Women Infected with SARS-CoV-2") examining aspects of immunology and microbiology in seven pregnant people with molecular evidence of SARS-CoV-2 compared to eight pregnant individuals without molecular evidence of SARS-CoV-2. The study examines immunological, gene expression, histological and microbiological/microbiome correlates of infection comparing to the uninfected control group.

These studies were conducted by a strong investigative team with experience in all of the methods conducted and a long track record of conducting research in reproductive immunology and microbiology. A major strength of this study is the interdisciplinary nature. A major weakness of the study is the very small number of subjects included.

Major comments:

1. The small number of largely asymptomatic SARS-CoV-2 positive patients limits inferences that can be drawn due to issues related to the uncertainty of when these patients got infected and/or whether these were initial infections or even reinfections. Obviously, it is not possible to draw conclusions about moderate/severe/symptomatic infection from these small case numbers. Despite some interesting differences between cases and controls, the biological meaning of these differences is difficult to infer given the very small number of study cases, the lack of information as to when these mothers were infected (or, for that matter, re-infected), the paucity of symptomatic patients, and a lack of adverse pregnancy outcomes (making it hard to know how important these findings might be in terms of understanding the risk of COVID-19-related adverse pregnancy outcomes).

2. The majority of subjects in the study (both infected and uninfected) were in labor at the time of enrollment. How was duration of labor incorporated into the analyses or possibly impact outlier datapoints? Similarly, how did mode of delivery (vaginal/section) impact results, including outliers?

3. The microbiome work, while somewhat interesting, does not seem to mesh well with the story / study as a whole and it is unclear how the data inform us about the impact of COVID-19 on microbial ecology, especially given that we do not know when these patients were infected.

Minor comments:

1. The authors should add to the discussion about limitations of the current study. There is little information about when subjects were infected with SARS-CoV-2, whether infected subjects knew

they were infected at any time prior to admission for delivery, how the presence or absence of labor was incorporated into the analyses, and the impact of small numbers of subjects and some missing data. Also, did the duration of labor differ or influence results? How were data adjusted for mode of delivery or presence/duration/treatment of labor?

2. On page 18, starting on line 372, the authors note that "maternal macrophage responses may act as a double-edged sword in the chorioamniotic membranes of women with SARS-CoV-2 infection by modulating host immune responses while simultaneously contributing to placental vasculopathy." This statement is highly speculative and cannot be concluded on the basis of the data presented in this study.

3. The y-axes are not labeled in figure 1B. Also, in figure 1C there appear to be missing data for IL-6, IL-17 and IFN-g in the control group, with only three data points each.

4. Cytokine levels in the results section include a lot of non-statistically significant trends, which is inappropriate.

5. The cytokines significantly upregulated in maternal circulation (IL-15) of infected individuals are different from the cytokines upregulated in the fetal cord blood (IL-17, TNF). Given that the fetus has no detectable IgM, it would be helpful to discuss potential mechanisms for the differential cytokine responses between uninfected fetuses and their respective infected mothers.

6. Line 329- the authors mention IL-6 levels in neonatal cord blood of infected asymptomatic mothers, please consider mentioning specifically the IL-6 levels in the cord blood from the infected symptomatic mothers as well.

7. Regarding immunophenotyping: it would be helpful if the authors could make information about the distinction between "macrophage-1" and "macrophage-2" and similarly "stromal-1" "stromal-2" and "stromal-3" easily available- what parameters were used to distinguish them?

8. If possible, specify in PCA plots which dot belongs to the severely infected patient/fetus of that patient similar to how this was done in the cytokine plots with a more saturated color.

9. Figure 2B, Figure 3: specify in text (results or discussion) that the lowest T cell counts in cord blood consistently (with the exception of cord blood Tc17, where it's the second-lowest) come from the severely infected patient

10. Figure 2C: mark with asterisk which column of infected patients belongs to the severely infected patient.

**Reviewer #1 (Remarks to the Author):**

The manuscript by Garcia-Flores et al describes the immune responses of pregnant women
that are SARS-CoV-2 infected at delivery and their placenta cell composition. The study is
performed in seven pregnant women that test positive for SARS-CoV2 at delivery and 8
normal pregnant women. Blood was collected from the mothers and from the umbilical cord
and shown analyzed for cytokines. It is unclear at this stage, whether those cytokines have
similar probabilities to cross the placenta and whether the ones found in the cord blood are of
embryonic origin. What appears clear though is that the T-cell lymphopenia observed in CoV
patients is not present in the cord blood T cells. Neither are the local placental fetal immune
responses. This is a carefully done study that shows that in mild cases of CoV 2 infection
(there is a single severe case in this study) the placenta is rarely infected, there is maternal
immune cell activation in the placenta but limited fetal immune reactions.

**Author response to overall comment:** We thank the Reviewer for their helpful feedback
and for taking the time to review our manuscript. We have addressed the Reviewer's
feedback below.

**Comment #1:** The authors have certainly data from neonatal T and B cells found in the
placenta. It would be interesting to show and discuss the transcriptional profile of these cells
compared to those from the mother and from control newborns.

**A. Author response to comment #1:** We thank the Reviewer for providing us with these
helpful recommendations that we believe have significantly improved our manuscript.
We have made substantial revisions to our manuscript, which are described in an
itemized manner below.

- a. We have collected a total of eight sets of new samples [five additional SARS-
CoV-2-infected pregnant women (two severe and three asymptomatic cases)
and three controls], which brings our total group sizes to 12 SARS-CoV-2 cases
(3 severe cases and 9 asymptomatic) and 11 controls. Each set of samples
included maternal blood, cord blood, and/or placental tissues. Experiments were
performed in all of these samples, which resulted in revisions to Figure 1
through 5, Figure 7, and the generation of a new Figure 6 and Supplementary
Figures 4, 7, and 9, and Supplementary Tables 4 through 8 and 12. Other
supplementary materials (Supplementary Figures 1, 2, 3, 5, 6, 8, and 10 and
Supplementary Tables 1, 2, and 9) were also revised.
- b. To specifically respond to this comment, we have performed bulk RNA-
sequencing of the cord blood and maternal blood samples from all women
included in this study from which such samples were available (cord blood:
SARS-CoV-2 (+) = 9, control = 8; maternal blood: SARS-CoV-2 (+) = 11, control
= 10). The results of this set of experiments are shown in the new Figure 6 and
Supplementary Figure 9 as well as Supplementary Tables 4 through 8. The
transcriptomes of the maternal blood and cord blood were significantly
correlated (new Figure 6b); however, some SARS-CoV-2 infection changes
were specific to the mother or the neonate (new Figure 6b-c). Gene Ontology
analysis revealed that the biological processes enriched in the upregulated
differentially expressed genes (DEGs) in maternal blood included activation of
the humoral immune response, including the classical pathway of complement

activation, adaptive immune responses, and immunoglobulin-mediated immune response, whereas the downregulated DEGs included phagocytosis and extracellular matrix organization (new Figure 6d). In contrast, the biological processes enriched in the upregulated DEGs in cord blood were associated with defense response to fungus and bacterium (new Figure 6e). No significant biological processes were enriched in the downregulated DEGs. These results show that SARS-CoV-2 infection alters shared and non-shared specific immune processes in the mother and offspring.

- c. Furthermore, as suggested by the Reviewer, our newly generated bulk RNA-sequencing data from cord blood and maternal blood (new Figure 6a-e) were intersected with the single-cell RNA-sequencing (RNA-seq) immune signatures that were altered by SARS-CoV-2 infection (Figure 4d). Notably, chorioamniotic membrane (CAM) maternally-derived scRNA-seq signatures of T cell, Macrophage-2, and Monocyte (non-significant) were correlated with the maternal blood transcriptome, and the placental villi and basal plate (PVBP) fetally-derived scRNA-seq T cell signature was correlated with the cord blood transcriptome (new Fig. 6g). These data show that the transcriptomic profile of the mother and the neonate correlate with the maternal and fetal immune responses in the placenta. Correlation of SARS-CoV-2 changes in bulk transcriptomic data and those of B cells in the placenta were not feasible due to the rarity of these cells.

B. Revisions located at:

- a. **Changes resulting from the addition of new patients:** Figures 1- 5, new Figure 6, Figure 7; new Supplementary Figures 4, 7, and 9; Supplementary Figures 1 - 3, 5, 6, 8, and 10; new Supplementary Tables 4 – 8 and 12; Supplementary Tables 1, 2, and 9; Results, Pages 6 - 8, 10 - 13; Discussion, Page 16, 17, 19, 20.
- b. **Changes resulting from new RNA-seq experiments:** New Figure 6; New Supplementary Figure 9 and Supplementary Tables 4-8; Results, Pages 11-13; Methods, Page 32 - 33; Figure Legends, Page 62-63

**Reviewer #2 (Remarks to the Author):**

This is an interesting and broad study by Garcia-Flores and colleagues (“Maternal-Fetal
Immune Responses in Pregnant Women Infected with SARS-CoV-2”) examining aspects of
immunology and microbiology in seven pregnant people with molecular evidence of SARS-
CoV-2 compared to eight pregnant individuals without molecular evidence of SARS-CoV-2.
The study examines immunological, gene expression, histological and
microbiological/microbiome correlates of infection comparing to the uninfected control group.

These studies were conducted by a strong investigative team with experience in all of the
methods conducted and a long track record of conducting research in reproductive
immunology and microbiology. A major strength of this study is the interdisciplinary nature. A
major weakness of the study is the very small number of subjects included.

**Author response to overall comment:** We thank the Reviewer for the helpful feedback and
for taking the time to review our manuscript. We have addressed the Reviewer’s concerns
regarding the small number of subjects included in this study by collecting a total of eight
additional sets of new samples [five SARS-CoV-2-infected pregnant women (two severe and
three asymptomatic cases) and three controls], which brings our total group sizes to 12
SARS-CoV-2 cases (3 severe cases and 9 asymptomatic) and 11 controls. Each set of
samples included maternal blood, cord blood, and/or placental tissues. Experiments were
performed in all of these samples, which resulted in revisions to Figure 1 through 5, Figure 7,
and the generation of a new Figure 6 and Supplementary Figures 4, 7, and 9, and
Supplementary Tables 4 through 8 and 12. Other supplementary materials (Supplementary
Figures 1, 2, 3, 5, 6, 8, and 10 and Supplementary Tables 1, 2, and 9) were also revised. We
consider that the inclusion of these additional cases and controls has strengthened the
findings of this study.

**Major comments:**

**Comment #1:** The small number of largely asymptomatic SARS-CoV-2 positive patients
limits inferences that can be drawn due to issues related to the uncertainty of when these
patients got infected and/or whether these were initial infections or even reinfections.
Obviously, it is not possible to draw conclusions about moderate/severe/symptomatic
infection from these small case numbers. Despite some interesting differences between
cases and controls, the biological meaning of these differences is difficult to infer given the
very small number of study cases, the lack of information as to when these mothers were
infected (or, for that matter, reinfected), the paucity of symptomatic patients, and a lack of
adverse pregnancy outcomes (making it hard to know how important these findings might be
in terms of understanding the risk of COVID-19-related adverse pregnancy outcomes).

**A. Author response to comment #1:** We thank the Reviewer for raising these important
points. We are responding to each of the Reviewer’s comments in an itemized manner
below:

- a. The prevalence of SARS-CoV-2 infection among pregnant women is low. After
the COVID-19 pandemic began, multiple hospitals, including our institution,
implemented universal screening for women admitted to Labor and Delivery
units showing that the positivity rate was less than 5% (Am J Obstet Gynecol
MFM. 2020 Nov;2(4):100226; Clin Infect Dis. 2021 Mar 1;72(5):869-872; Am J

Perinatol. 2020 Sep;37(11):1110-1114; J Perinat Med. 2021 Jun 10;49(6):717-
722). Moreover, only a fraction of such women presented with severe COVID-19
(Am J Obstet Gynecol. 2021 Jul;225(1):77.e1-77.e14). Therefore, the
recruitment of women with SARS-CoV-2 infection, particularly those with severe
COVID-19, is extremely challenging. Nonetheless, our team was able to recruit
a total of five additional infected pregnant women, including two severe cases
and three asymptomatic cases, as well as three additional healthy controls,
which resulted in the addition of a new author and the reorganization of
authorship. We consider that the inclusion of these additional samples has
greatly strengthened our study.

- b. The infected patients included in this cross-sectional study were diagnosed with
SARS-CoV-2 at the time of admission using a PCR test. Given that the majority
of infected patients displayed high levels of IgM in the maternal circulation, it is
possible that these women were in the acute phase of infection at admission.
Regardless, we are now including a limitations section in the discussion noting
that the time of infection could not be considered for the interpretation of the
findings in the current study.
- c. As noted by the Reviewer, the majority of our SARS-CoV-2-infected patients did
not present short-term adverse pregnancy outcomes; yet, one of the newly-
included women with severe COVID-19 underwent emergency preterm
cesarean section due to worsening respiratory function. This finding coincides
with previous studies reporting that COVID-19 is associated with higher rates of
indicated preterm birth (Am J Obstet Gynecol. 2021 Jul;225(1):77.e1-77.e14).
Therefore, we consider that the findings reported in our study are timely.
Furthermore, the data generated in our study show that, even in the absence of
symptoms, neonates born to women infected with SARS-CoV-2 display aberrant
immune responses in the placenta and cord blood. Thus, our findings
underscore the potential long-term neonatal/infant consequences of SARS-CoV-
2 infection during pregnancy, even in asymptomatic cases, which is also
included in the revised discussion.

B. Revisions located at:

- a. **Changes resulting from the addition of new patients:** Figures 1- 5, new
Figure 6, Figure 7; new Supplementary Figures 4, 7, and 9; Supplementary
Figures 1 - 3, 5, 6, 8, and 10; new Supplementary Tables 4 – 8 and 12;
Supplementary Tables 1, 2, and 9; Results, Pages 6 - 8, 10 - 13; Discussion,
Page 16, 17, 19, 20.
- b. **New study limitations section:** Discussion, Page 21

**Comment #2:** The majority of subjects in the study (both infected and uninfected) were in
labor at the time of enrollment. How was duration of labor incorporated into the analyses or
possibly impact outlier datapoints? Similarly, how did mode of delivery (vaginal/section)
impact results, including outliers?

**A. Author response to comment #2:** We thank the Reviewer for pointing out this matter,
which we are responding to in an itemized manner.

- a. The presence of labor and rate of cesarean section were both similar between
the study and control groups, as shown in Supplementary Table 1. Regardless,
we performed a model sensitivity analysis to determine whether adding these

two variables and additional covariates in the DESeq2 linear model could have
a significant impact on any reported differences between the study groups.

- b. For scRNA-seq analyses, we determined that adding a term controlling for
library preparation batch in the model yielded the best results in terms of
number of differentially expressed genes. We also evaluated the contribution of
additional covariates (labor and delivery route); however, their impact was
minimal compared to the model adjusting for batch effects only. Therefore,
results after adjustment for batch effect are reported. We also kindly ask the
Reviewer to consider that we lacked the statistical power to evaluate the effects
of additional covariates in the model utilized to analyze scRNA-seq data. The
adjustments performed in scRNA-seq data are now mentioned in the revised
methods section.
- c. With the newly added bulk RNA-seq data, in which all libraries were prepared in
one batch, the batch effect was minimal. Yet, the number of samples allowed us
to use a model that included maternal age, BMI, nulliparity, labor status, and
delivery route as covariates. The adjustments performed in the bulk RNA-seq
data are now mentioned in the revised methods section.

**B. Revisions located at: Methods, Pages 30, 33**

**Comment #3:** The microbiome work, while somewhat interesting, does not seem to mesh
well with the story / study as a whole and it is unclear how the data inform us about the impact
of COVID-19 on microbial ecology, especially given that we do not know when these patients
were infected.

**A. Author response to comment #3:** We thank the Reviewer for requesting this
clarification. We apologize for not justifying well the reasoning behind our investigation
of the placental microbiome in the current study. The following paragraph has been
added to the results section:

*“The traditional view is that the placenta is a sterile organ that is first colonized by*
*vaginal microbes during delivery.^{39,40} However, the sterility of the placenta could be*
*compromised by microorganisms invading from the lower genital tract (i.e., ascending*
*infection) or those present in the maternal circulation (i.e., hematogenous infection).^{41,42}*
*Therefore, we evaluated whether infection with the SARS-CoV-2 virus, which can be*
*detected in vaginal fluid¹⁵ or the peripheral circulation,⁴³ could compromise the sterility*
*of the placenta.”*

We have now further justified the inclusion of these experiments/data in our study.

**B. Revisions located at: Results, Page 15**

**Minor Comments:**

**Comment #4:** The authors should add to the discussion about limitations of the current study.
There is little information about when subjects were infected with SARS-CoV-2, whether
infected subjects knew they were infected at any time prior to admission for delivery, how the
presence or absence of labor was incorporated into the analyses, and the impact of small
numbers of subjects and some missing data. Also, did the duration of labor differ or influence
results? How were data adjusted for mode of delivery or presence/duration/treatment of
labor?

A. Author response to comment #4: We thank the Reviewer for bringing up these concerns. We are responding to the Reviewer in an itemized manner below:

- a. We are now including a limitations section in the discussion noting that the time of infection could not be considered in the current study, and acknowledging the number of samples included in the study.
- b. We performed additional analyses to determine whether covariates such as the presence of labor and mode of delivery could have a significant impact on the differences in single-cell transcriptomic data between the study groups. We performed a model sensitivity analysis to determine whether adding labor and delivery route as covariates could have a significant impact on any reported differences between the study groups. All scRNA-seq models evaluated included a batch variable; adding labor or mode of delivery only minimally increased the number of genes detected. Therefore, the scRNA-seq results presented in the current study were only adjusted for library preparation batch. Yet, the model utilized to analyze newly generated bulk RNA-seq data included maternal age, BMI, nulliparity, labor status, and delivery route as covariates.

B. Revisions located at: Discussion, Page 21; Methods, Pages 30, 33

Comment #5: On page 18, starting on line 372, the authors note that “maternal macrophage responses may act as a double-edged sword in the chorioamniotic membranes of women with SARS-CoV-2 infection by modulating host immune responses while simultaneously contributing to placental vasculopathy.” This statement is highly speculative and cannot be concluded on the basis of the data presented in this study.

A. Author response to comment #5: We thank the Reviewer for this comment. We have edited the above mentioned statement in the revised manuscript.

B. Revisions located at: Discussion, Page 19

Comment #6: The y-axes are not labeled in figure 1B. Also, in figure 1C there appear to be missing data for IL-6, IL-17 and IFN-g in the control group, with only three data points each.

A. Author response to comment #6: We thank the Reviewer for bringing up these points. We have addressed each of these concerns in an itemized manner below:

- a. The y-axes have been revised to include the proper labels. We apologize for this oversight.
- b. The cytokine data shown in Figure 1 were reanalyzed based on the Reviewer’s comment. In the revised Figure 1b-c and Supplementary Figures 1-2, the geometric mean was used to summarize data from duplicates to attenuate the effect of outlier values. If only one of the duplicate values was below the detection limit, the value above the detection limit was taken for that patient. Data below the detection limit in both duplicates were imputed with 99% of the minimum detected value across any sample. Differences between groups were assessed by linear mixed-effects models after log-transformation of the data. Therefore, cases were weighted differently depending on the number of data points above the detection limit and the within subject variance. The revised Supplementary Table 2 also provides the log₂ fold changes for each cytokine.

The revised analysis is now included in the revised methods section (Statistical
analysis).

**B. Revisions located at:** Figure 1b-c, Supplementary Figures 1-2; Methods, Page 41

**Comment #7:** Cytokine levels in the results section include a lot of non-statistically significant
trends, which is inappropriate.

**A. Author response to comment #7:** We thank the Reviewer for pointing out this matter.
We have revised the cytokine results to only describe those findings that were
statistically significant.

**B. Revisions located at:** Results, Pages 7-8

**Comment #8:** The cytokines significantly upregulated in maternal circulation (IL-15) of
infected individuals are different from the cytokines upregulated in the fetal cord blood (IL-17,
TNF). Given that the fetus has no detectable IgM, it would be helpful to discuss potential
mechanisms for the differential cytokine responses between uninfected fetuses and their
respective infected mothers.

**A. Author response to comment #8:** We thank the Reviewer for raising this point. After
the inclusion of additional data from the new SARS-CoV-2-infected and control patients,
we no longer observe changes in IL-17A or TNF in the cord blood (revised Figure 1c).
Now, the only cytokine increased in the cord blood of neonates born to SARS-CoV-2-
infected women is IL-8, which is also observed in the maternal circulation. This finding
may represent the transfer of maternal cytokines through the placental tissues, which
was mentioned in the discussion section.

**B. Revisions located at:** Figure 1b-c; Supplementary Figures 1-2; Results, Pages 7-8;
Discussion, Pages 16-17

**Comment #9:** Line 329- the authors mention IL-6 levels in neonatal cord blood of infected
asymptomatic mothers, please consider mentioning specifically the IL-6 levels in the cord
blood from the infected symptomatic mothers as well.

**A. Author response to comment #9:** We thank the Reviewer for bringing up this point.
After the addition of new cases and controls, no significant changes in IL-6
concentrations were observed in the cord blood of SARS-CoV-2 cases. Therefore, we
prefer to focus on the discussion of newly generated data.

**B. Revisions located at:** N/A

**Comment #10:** Regarding immunophenotyping: it would be helpful if the authors could make
information about the distinction between “macrophage-1” and “macrophage-2” and similarly
“stromal-1” “stromal-2” and “stromal-3” easily available- what parameters were used to
distinguish them?

**A. Author response to comment #10:** We thank the Reviewer for requesting this
clarification. The cell populations utilized in this study were defined according to
previously published single-cell RNA-sequencing marker genes, as described in our
prior study (Elife. 2019 Dec 12;8:e52004). Yet, the list of genes utilized for macrophage
and stromal cell clusters is now shown as Supplementary Table 12.

**B. Revisions located at:** Supplementary Table 12; Methods, Page 29

**Comment #11:** If possible, specify in PCA plots which dot belongs to the severely infected
patient/fetus of that patient similar to how this was done in the cytokine plots with a more
saturated color.

**A. Author response to comment #11:** We thank the Reviewer for the kind
recommendation. We have modified all plots, including PCA plots, accordingly to
denote the samples derived from severely infected patients.

**B. Revisions located at:** Figures 1, 2, 3, 7, and 8; Supplementary Figures 1 - 5, and 10;
Figure Legends, Pages 59 - 65

**Comment #12:** Figure 2B, Figure 3: specify in text (results or discussion) that the lowest T
cell counts in cord blood consistently (with the exception of cord blood Tc17, where it's the
second-lowest) come from the severely infected patient.

**A. Author response to comment #12:** We thank the Reviewer for pointing out this
finding. We have included five new SARS-CoV-2-infected cases in our analysis, and
after the addition of these samples the above observation is no longer reported.

**B. Revisions located at:** Figure 2b, Figure 3c

**Comment #13:** Figure 2C: mark with asterisk which column of infected patients belongs to
the severely infected patient.

**A. Author response to comment #13:** We thank the Reviewer for the helpful
recommendation. We have provided color coding to indicate the controls (blue),
asymptomatic infected patients (light red), and the severe infected cases (dark red).

**B. Revisions located at:** Figure 2c and Supplementary Figures 3-4; Figure Legends,
Page 59-60

Reviewers' Comments:

Reviewer #1:

Remarks to the Author:

The authors addressed my main concerns. The manuscript significantly increased in quality.

Reviewer #2:

Remarks to the Author:

The authors have addressed my concerns and improved the quality of the manuscript considerably.